# Single-cell RNA-seq enables comprehensive tumour and immune cell profiling in primary breast cancer

Woosung Chung[1,2,*], Hye Hyeon Eum[1,3,*], Hae-Ock Lee[1,4,*], Kyung-Min Lee[5,6], Han-Byoel Lee[5,7], Kyu-Tae Kim[1], Han Suk Ryu[8], Sangmin Kim[9], Jeong Eon Lee[9], Yeon Hee Park[10], Zhengyan Kan[11], Wonshik Han[5,7] & Woong-Yang Park[1,2,4]

Single-cell transcriptome profiling of tumour tissue isolates allows the characterization of heterogeneous tumour cells along with neighbouring stromal and immune cells. Here we adopt this powerful approach to breast cancer and analyse 515 cells from 11 patients. Inferred copy number variations from the single-cell RNA-seq data separate carcinoma cells from non-cancer cells. At a single-cell resolution, carcinoma cells display common signatures within the tumour as well as intratumoral heterogeneity regarding breast cancer subtype and crucial cancer-related pathways. Most of the non-cancer cells are immune cells, with three distinct clusters of T lymphocytes, B lymphocytes and macrophages. T lymphocytes and macrophages both display immunosuppressive characteristics: T cells with a regulatory or an exhausted phenotype and macrophages with an M2 phenotype. These results illustrate that the breast cancer transcriptome has a wide range of intratumoral heterogeneity, which is shaped by the tumour cells and immune cells in the surrounding microenvironment.

[1] Samsung Genome Institute, Samsung Medical Center, Seoul 06351, Korea. [2] Department of Health Sciences and Technology, Samsung Advanced Institute for Health Sciences & Technology, Sungkyunkwan University, Seoul 06351, Korea. [3] Department of Biomedical Sciences, Seoul National University Graduate School, Seoul 03080, Korea. [4] Department of Molecular Cell Biology, Sungkyunkwan University School of Medicine, Suwon 16419, Korea. [5] Department of Surgery and Cancer Research Institute, Seoul National University College of Medicine, Seoul 03080, Korea. [6] Biomedical Research Institute, Seoul National University Hospital, Seoul 03080, Korea. [7] Department of Surgery, Seoul National University College of Medicine, Seoul 03080, Korea. [8] Department of Pathology, Seoul National University College of Medicine, Seoul 03080, South Korea. [9] Department of Surgery, Samsung Medical Center, Sungkyunkwan University School of Medicine, Seoul 06351, Korea. [10] Division of Hematology-Oncology, Department of Medicine, Samsung Medical Center, Seoul 06351, Korea. [11] Oncology Research, Pfizer Inc., San Diego, California 92121, USA. * These authors contributed equally to this work. Correspondence and requests for materials should be addressed to W.H. (email: hanw@snu.ac.kr) or to W.-Y.P. (email: woongyang@skku.edu).

Many molecular-targeted treatments for breast cancer have been evaluated since the application of endocrine therapy for oestrogen receptor (ER)-positive tumour types[1]. Genome alteration-matched treatment of breast cancer to target amplification of human epidermal growth factor receptor 2 gene (Erb-B2 receptor tyrosine kinase 2, *ERBB2* also known as *HER2*) is an example of a successful gene-targeted therapy[2]. Gene expression-based molecular subtyping has also been broadly applied to breast cancer to aid treatment decisions[3,4]. Molecular targeted approaches have broadened the treatment options for breast cancer and have significantly improved therapeutic outcomes. However, genomic and gene expression profiling are usually used to characterize a bulk tumour in individual cancer patients, whereas cancers display intratumoral heterogeneity that might affect the therapeutic outcome of a targeted treatment.

Genetic heterogeneity in breast cancer has been demonstrated at a single-cell resolution with high levels of genome coverage[5]. Copy number alterations were found at the early stage of cancer development and remained stable, whereas single-nucleotide variations varied extensively throughout tumour evolution. A recent report on genetic alterations in HER2-negative regions among HER2-amplified backgrounds[6] demonstrated multiple driver mutations in a single tumour, suggesting a direct influence of genetic heterogeneity on the therapeutic outcome. Heterogeneity at the level of gene expression also greatly influences the clinical outcome of patients. Approximately 20% of ER+ tumours either do not respond to endocrine therapy or develop acquired treatment resistance at a later stage. The presence of ER-negative cells in an ER+ tumour represents a molecular mechanism of drug resistance. Rewired signalling pathway activation through other growth factor receptors, as well as a change in the subtype from that of the primary tumour in metastatic lesions[7], may also arise from gene expression level heterogeneity in individual tumour cells.

Gene expression profiling in bulk tumours reflects the features of non-tumour compartments, which, in the case of breast cancer, are characterized by an extensive admixture of stromal, immune and endothelial cell infiltration. These admixtures form the tumour microenvironment and play a critical role in tumour initiation, progression and treatment resistance. Micro-environmental gene expression signatures may themselves present prognostic values independent of the intrinsic tumour subtype[8–10]. The major cell populations forming the cancer microenvironment include cancer-associated fibroblasts[11] and immune cells[12]. Immune cell infiltrates are composed of cells from multiple lineages that may play pleiotropic roles. Tumour-associated macrophages (TAMs) often promote tumour progression and metastasis, whereas CD8+ cytotoxic T cells and CD4+ Th1 cells elicit antitumour immunity and suppress tumour growth[13]. Furthermore, T cells with regulatory or exhausted phenotype are associated with failure in antitumour immunity. A subset of B cells was proposed to promote tumour progression by affecting diverse cell types including T cells and TAMs[14]. However, the presence of a large number of B cells in the tumour region is associated with a good prognosis[15]. Altogether, the tumour microenvironment is formed through interactions between these variable cellular components and through communication with tumour cells.

Single-cell genome analysis is expected to have clinical utility in cancer treatment[16]. Non-invasive monitoring of circulating tumour cells, estimation of tumour heterogeneity, early detection of small numbers of recurrent tumours and sensitive monitoring of rare cell populations can be utilized for patient care in cases of refractory cancers. Knowing the level of transcriptome heterogeneity in the tumour entity and the precise characterization of tumour and microenvironmental gene expression may help identify better molecular targets for prognosis and treatment[17]. Characterization of heterogeneous tumour signatures will lead to effective molecular targeted therapies, whereas characterization of tumour-infiltrating immune cells may reveal a better strategy for overcoming immune suppression and revitalizing the naturally occurring immune surveillance[18].

We provide the transcriptome analysis of 515 single cells from 11 patients with different breast cancer subtypes. Single-cell isolates from individual tumour tissues contain carcinoma and non-carcinoma microenvironment cells. Each population has a unique pattern of gene expression, which cannot be resolved in the total mixed population. Taken together, our study reveals the characteristics of different tumour subtypes that are shaped by tumour cells and immune cells in the microenvironment.

## Results

**Genomic profiles of 11 patients for single-cell analysis**. We selected 11 patients representing the four subtypes of breast cancer: luminal A; luminal B; HER2; and triple negative breast cancer (TNBC). All but one of the surgical samples were obtained from chemotherapy-naive patients, and the markers for subtyping were validated by pathological examination as ER-positive (BC01 and BC02; luminal A), ER/HER2-positive (BC03; luminal B), HER2-positive (BC04, BC05 and BC06; HER2) and triple-negative (BC07–BC11; TNBC) invasive ductal carcinoma (Supplementary Table 1). Regional metastatic lymph nodes were collected from the luminal B (BC03LN) sample and a triple-negative breast cancer (BC07LN) sample.

Whole-exome sequencing revealed the genomic landscape of these samples regarding somatic mutations (Supplementary Data 1) and copy number variations (CNVs; Supplementary Data 2). Triple-negative breast cancer patients showed marked alterations of CNVs (Supplementary Data 2), supporting previous reports on extensive genomic instability in this subtype tumour[19]. We also confirmed the characteristic genetic alterations in breast cancer, including missense mutations or amplifications in *PIK3CA* (4/11 patients), missense mutations or deletions in *TP53* (5/11 patients) and amplifications in *ERBB2* (4/11 patients; Supplementary Fig. 1)[20–22].

We isolated single cells using microfluidic chips[23] without prior cell type selection to generate RNA-seq data containing $5.8 \pm 1.3$ million reads from the amplified cDNAs of each single cell (Supplementary Data 3). Detection of constant ratios of two spiked-in RNAs assured the quality and consistency of all single-cell RNA-seq experiments (Supplementary Fig. 2a). Quantitative PCR analysis of the expression of 24 selected genes supported the data from single-cell RNA-seq (Supplementary Fig. 2b). Pooled tissue isolates were highly reflective of the matching tumour tissues (Supplementary Fig. 2c). Comparisons between the averages of single cells and corresponding pooled samples (Supplementary Fig. 2d) demonstrated partial but significant correlations (Pearson's $r$ 0.16–0.63 with average 0.47, $P < 0.001$). Multiple regression analyses of the transcriptomes of different-sized pools of single cells to those of bulk tumours (Supplementary Fig. 2e) provided a better representation of the tumour population with an increasing number of single cells. Altogether, single-cell RNA-seq could illustrate a significant portion of the tumour entity, yet tumour components were lost during the single-cell isolation or sequencing processes.

**Separation of tumour and tumour-associated normal cells**. We analysed the gene expression profiles of 515 tumour tissue isolates and found extensive intratumoral heterogeneity, as shown by the mixed representation of intra- and inter-patient cells by principal component analysis (Fig. 1a). Expression of therapeutic target

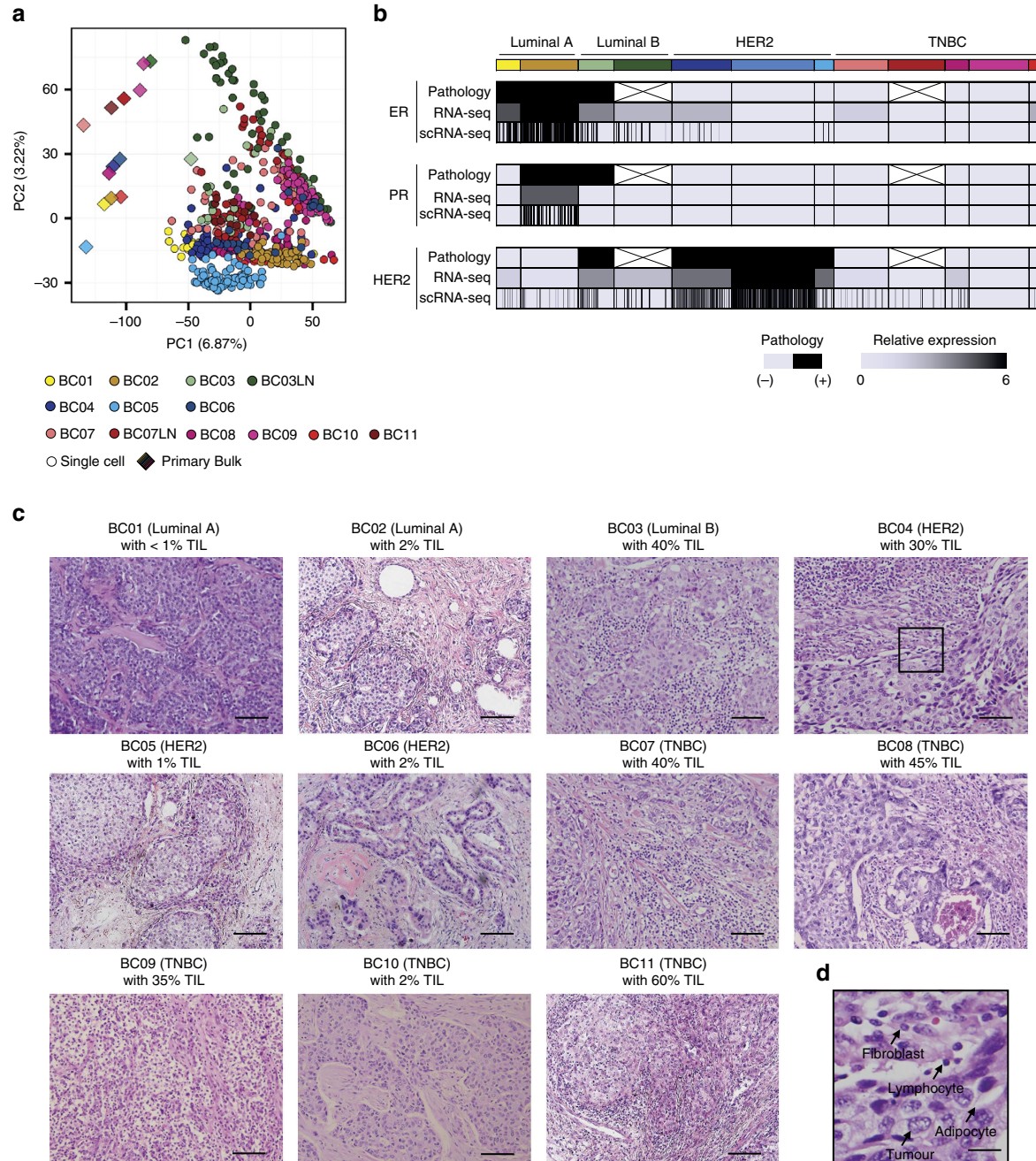

**Figure 1 | Intratumoral heterogeneity in primary breast tumours. (a)** Unsupervised PCA on the transcriptome, indicating a mixed distribution of intra- and interpatient cells. Individual cells are coloured yellow for luminal A, green for luminal B, blue for HER2, and red for TNBC tumours. This colour scheme is maintained throughout the manuscript. **(b)** Individual cells exhibiting gene expression heterogeneity for ER (*ESR1*), PR (*PGR*) and HER2 (*ERBB2*). The overall single-cell expression profiles agree with the bulk tumour expression profiles and the pathology results. **(c)** Haematoxylin and eosin staining on formalin-fixed paraffin-embedded slides. Microscopic findings indicated carcinoma and non-carcinoma cells, including tumour-infiltrating lymphocytes[9] (TIL, 1–60%). Most of the TNBC tumours except BC10 were heavily infiltrated with lymphocytes, whereas luminal A tumours showed enrichment with carcinoma cells. Scale bar, 100 μm. **(d)** A part of the tumour tissue in **c** is magnified to show non-neoplastic cellular components as a representative. Scale bar, 25 μm.

genes such as ER (*ESR1*) and HER2 (*ERBB2*) also varied across and within tumours (Fig. 1b). Because we isolated single cells without prior selection of carcinoma cells, we speculated that non-carcinoma cells might contribute to the observed intratumoral heterogeneity. Those non-carcinoma populations potentially represent fibroblasts, adipocytes, endothelial cells and diverse immune cells according to the histopathological examinations of the tumour tissues (Fig. 1c,d). Each tumour manifested differential level of immune cell infiltration such that luminal A type (BC01 and BC02) tumours were highly enriched with carcinoma cells, whereas most TNBC type (BC07–10) tumours showed extensive immune cell infiltration[9] (Fig. 1c).

Gene expression profiling of tumour tissues reflects the signatures of both the tumour and the surrounding microenvironment. Therefore, single-cell transcriptome profiling enables the separation of tumour-specific gene expression

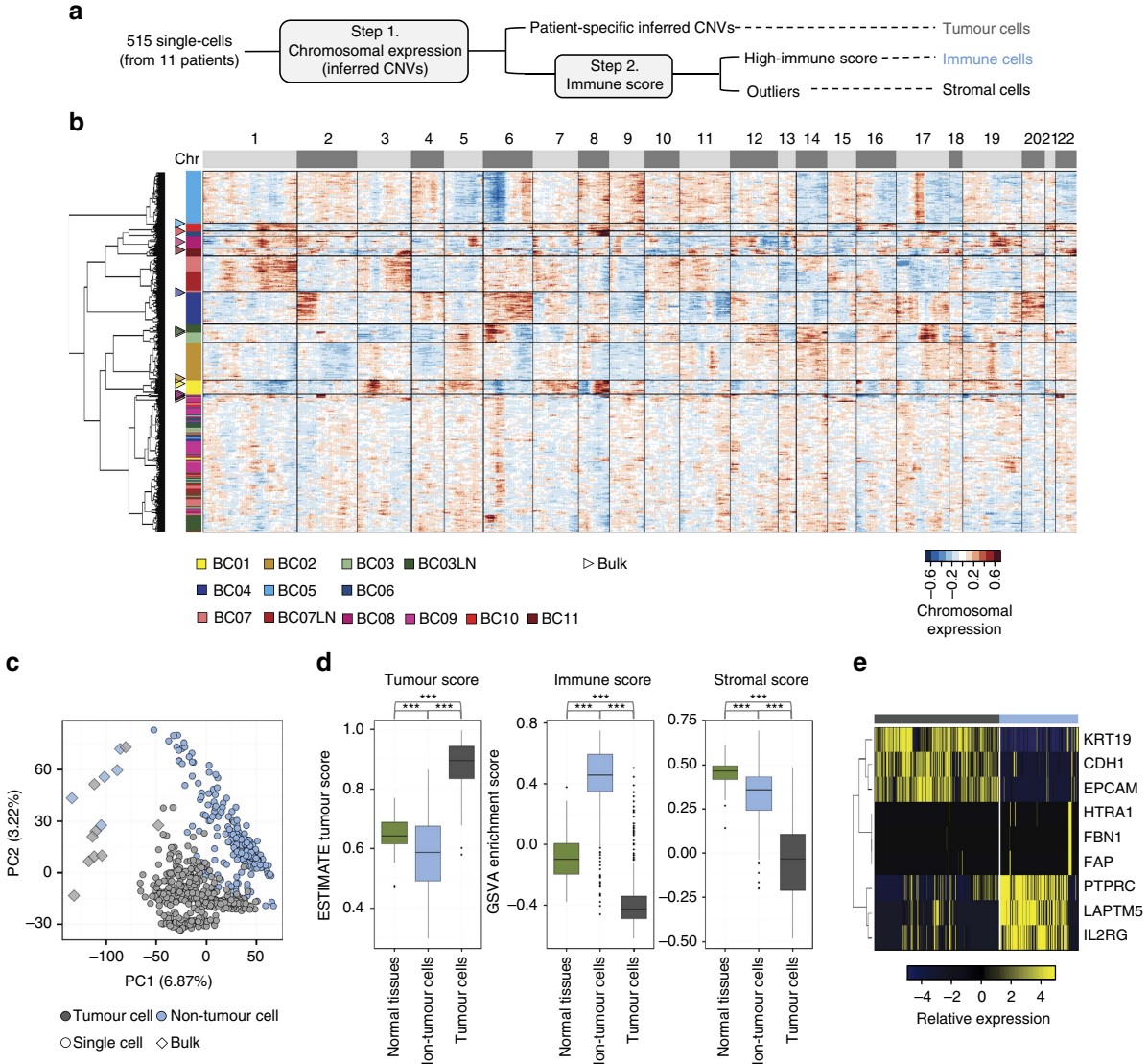

**Figure 2 | Separation of carcinoma and non-carcinoma cells. (a)** Scheme of cell classification. **(b)** Hierarchical clustering of the chromosomal gene expression pattern separating the patient-specific carcinoma cell groups from the non-carcinoma cell cluster. Each row represents single cells and matched bulk tumours (triangle): the tumour groups are colour-coded as in Fig. 1a. For each chromosome, the chromosomal gene expression pattern was estimated from the moving average of 150 genes. These patterns implicate chromosomal amplification and deletion. **(c)** Unsupervised PCA showing the separation of carcinoma and non-carcinoma cell groups. **(d)** Carcinoma cells identified in **a**, scored low for stromal and immune signatures, whereas non-carcinoma cells scored high for immune signatures. Tumour score was inferred from the stromal and immune signature using ESTIMATE algorithm[26]. Normal tissues represent 183 mammary tissue data from GTEx portal (http://www.gtexportal.org/). Each box shows the median and interquartile range (IQR 25th–75th percentiles), whiskers indicate the highest and lowest value within 1.5 times the IQR and outliers are marked as dots. *P* value, Student's *t*-test (\*\*\**P* < 0.001). **(e)** Representative gene expression in single cells for the immune (*PTPRC*, *LAPTM5* and *IL2RG*), stromal (*HTRA1*, *FBN1* and *FAP*) and epithelial (*KRT19*, *CDH1* and *EPCAM*) cell types.

signatures from those of non-tumour compartments. We used the chromosomal gene expression patterns to separate the breast carcinoma cells from the mixed population. Here gene expression profiles were aligned along the chromosomes as moving averages[24,25]. With this approach, tumour cells would have distinct chromosomal expression patterns, recapitulating tumour-specific CNVs[24] (Fig. 2a). Unsupervised hierarchical clustering of chromosomal gene expression patterns identified 11 clusters (Fig. 2b) among the 515 single cells: patient-specific carcinoma clusters with distinct chromosomal gene expression patterns and a multi-patient non-carcinoma cell cluster with no apparent CNV patterns. The chromosomal gene expression pattern in carcinoma clusters largely recapitulated the genomic CNV

profiles (Supplementary Fig. 3). The separation of carcinoma versus non-carcinoma cells was corroborated by the results of the unsupervised principal component analysis of the transcriptome (Fig. 2c).

To further delineate the identity of carcinoma and non-carcinoma cells, we analysed the expression of tumour-associated stromal or immune gene sets proposed in the tumour purity estimation, ESTIMATE[26]. Most of the non-carcinoma cells scored high for the immune signature (Fig. 2d). A number of non-carcinoma cells with low 'immune scores' expressed stromal genes (Fig. 2d,e), suggesting their identity as cancer-associated fibroblasts. Most carcinoma populations scored low for both stromal and immune gene expression, while expressing high

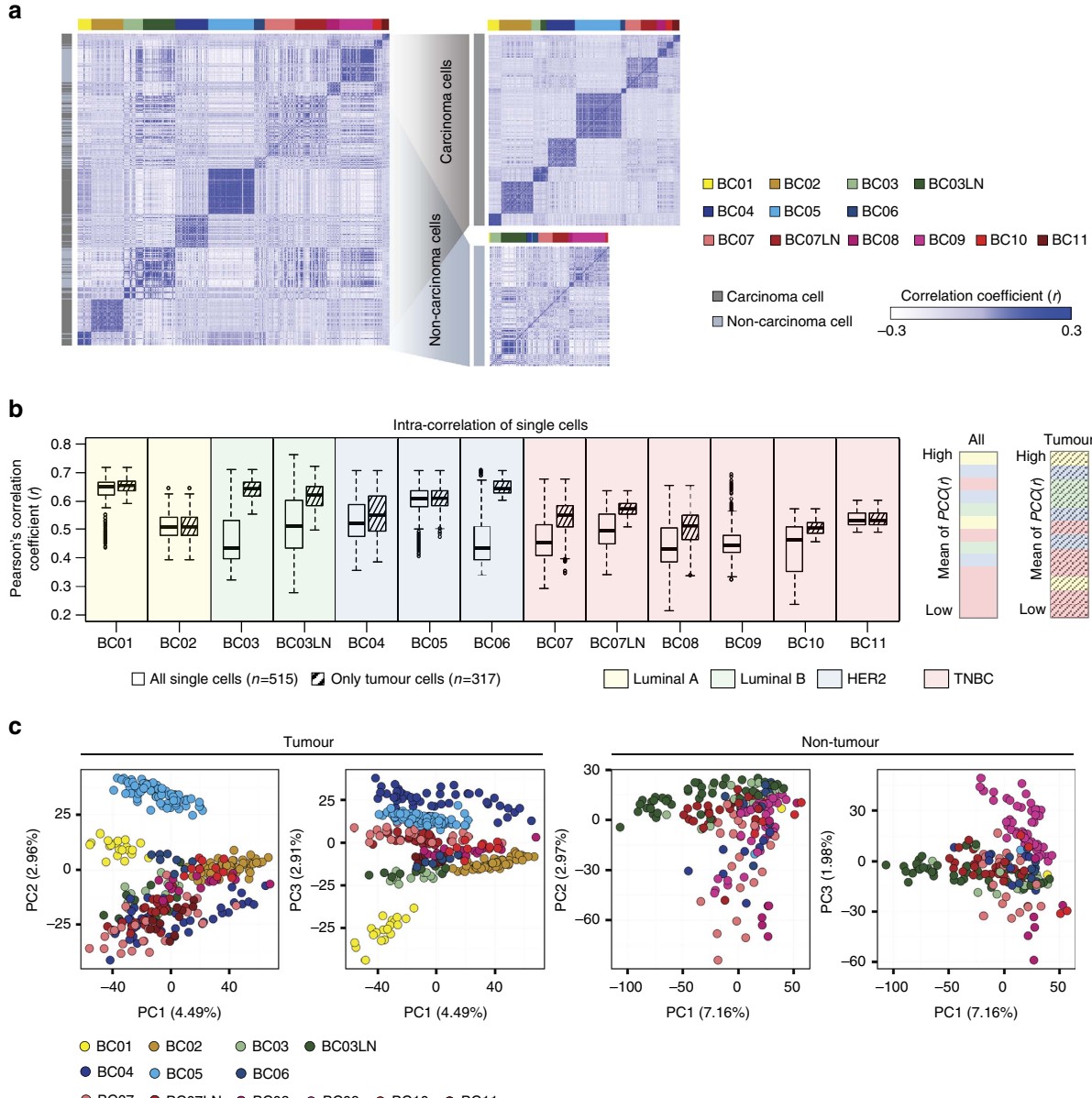

**Figure 3 | Removal of non-carcinoma cells reveals intrinsic tumour cell heterogeneity.** (**a**) Centred correlation matrix for all single cells demonstrates low cell-to-cell correlations (Pearson's *r*) in tumours with lymph node metastases (left). After separation of carcinoma and non-carcinoma cells, cell-to-cell correlations within the same tumour group are increased (right). Each row and column represents single cells. In the colour panel on the far left side, grey represents tumour and light blue represents non-tumour cells. (**b**) Intratumoral correlations before (white boxes) and after (striped boxes) the removal of non-carcinoma cells (left). Each box shows the median and interquartile range (IQR 25th–75th percentiles), whiskers indicate the highest and lowest value within 1.5 times the IQR and outliers are marked as dots. Samples were ranked by mean value of cell-to-cell Pearson's correlation coefficient (right). (**c**) Unsupervised PCA on the transcriptome separating patient-specific tumour groups for only tumour cells.

levels of epithelial cell differentiation markers. Altogether, among 515 single cells, we estimated the capture of 317 epithelial breast cancer cells, 175 tumour-associated immune cells and 23 non-carcinoma stromal cells (Fig. 2a).

**Intrinsic tumour cell heterogeneity in cancer aggressiveness.** Analysis of the cell-to-cell correlations for gene expression of the 515 cells demonstrated a relatively low degree of similarity between cells from the same patient (Fig. 3a). After excluding 198 non-carcinoma cells, we found a marked increase in the correlation between single carcinoma cells within a patient, suggesting that intrinsic tumour cell properties and heterogeneity

should be resolved at the single-cell level. Carcinoma cells from TNBC-type tumours (BC07–11) tend to show low cell-to-cell correlations with or without the removal of non-carcinoma cells (Fig. 3b), suggesting the contribution of both tumour intrinsic and microenvironmental properties to the intratumoural heterogeneity in TNBC. Ultimately, single-cell analysis revealed the distinct carcinoma characteristics of each patient tumour and diverse microenvironmental populations shared by different patient tumours (Fig. 3c).

The putative breast cancer cells were further classified into molecular subgroups using joint distribution between the ER and HER2 module scores, scaling the ER and HER2 signalling, respectively[27,28]. For the molecular subgroup analysis,

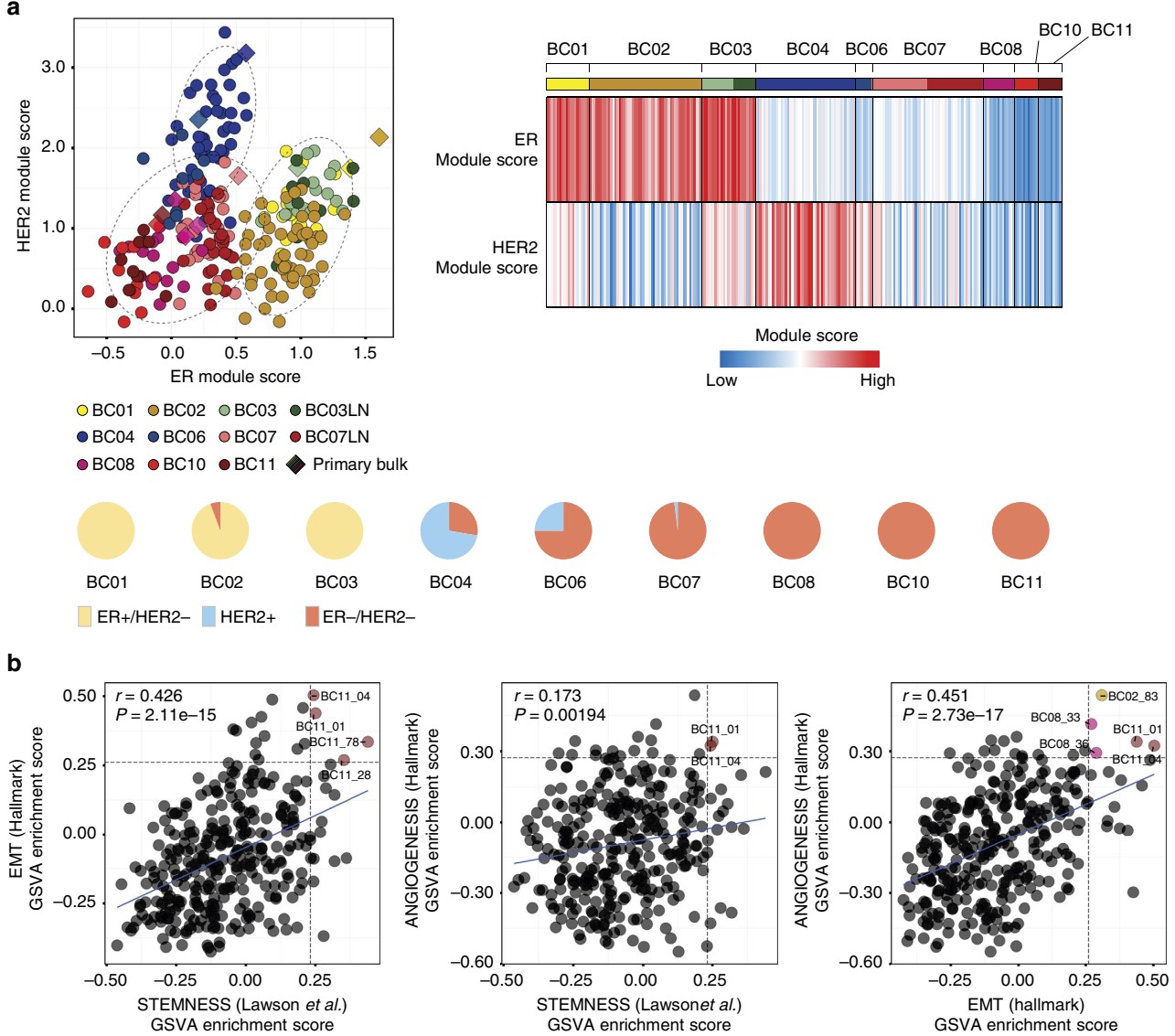

**Figure 4 | Intrinsic tumour cell heterogeneity.** (**a**) Subtype prediction of tumour cells as ER+, HER2+ or TNBC types using ER and HER2 module scores[50] (R software package genefu, top left panel). Heatmap of subtype-specifying module scores (top right) and pie charts (bottom) demonstrating Intratumoral subtype heterogeneity in HER2 tumours. (**b**) Pearson's correlation coefficient (r) between aggressive cancer gene expression signatures (EMT, stemness and angiogenesis). Horizontal and vertical dashed lines indicate signature score cut-offs at top 5%. The linear regression result is drawn as a solid line. The coloured dots mark cells with high expression signatures for both axis.

we excluded BC05 tumour cells, which had been subjected to neoadjuvant chemotherapy and Herceptin. Application of this classification method predicted the subtype of The Cancer Genome Atlas (TCGA) breast cancer data set with 91% accuracy (Supplementary Fig. 4). Most carcinoma cells were confined within the boundary of the corresponding subtype category of the bulk tumour (Fig. 4a, upper left), whereas some tumours show mixed subtype cells (Fig. 4a, upper right). In particular, carcinoma cells from HER2-positive tumours had the most variable subtype composition, from predominant HER2 to mostly TNBC types (Fig. 4a, bottom). Carcinoma cells from the ER/HER2 double-positive tumour (BC03) were categorized as the ER subtype with concomitant but low expression of HER2 module genes.

Carcinoma cells variably expressed aggressive cancer signatures such as epithelial–mesenchymal transition (EMT), stemness, angiogenesis, proliferation[29,30] and recurrence[3,31] (Supplementary Fig. 5a). Overall, the TNBC-type carcinoma cells had higher expression of EMT signatures than those from the luminal and HER2 types (Supplementary Fig. 5b). Both HER2 and TNBC tumour cells expressed high levels of stemness and recurrence signatures. Notably, EMT, stemness and angiogenesis signatures had positive correlations (Fig. 4b), and rare cells with concurrent high expression were identified. These rare subpopulations may play a critical role in tumour cell propagation, progression and metastasis.

**Heterogeneity components in breast cancer subtypes.** Most carcinoma cells had gene expression characteristics consistent with the parental group after the removal of non-carcinoma cells (Fig. 3c). Using the single-cell data set, we identified differentially expressed genes between breast cancer subtypes. We applied the R package Seurat[32] and used the likelihood ratio test (LRT) based

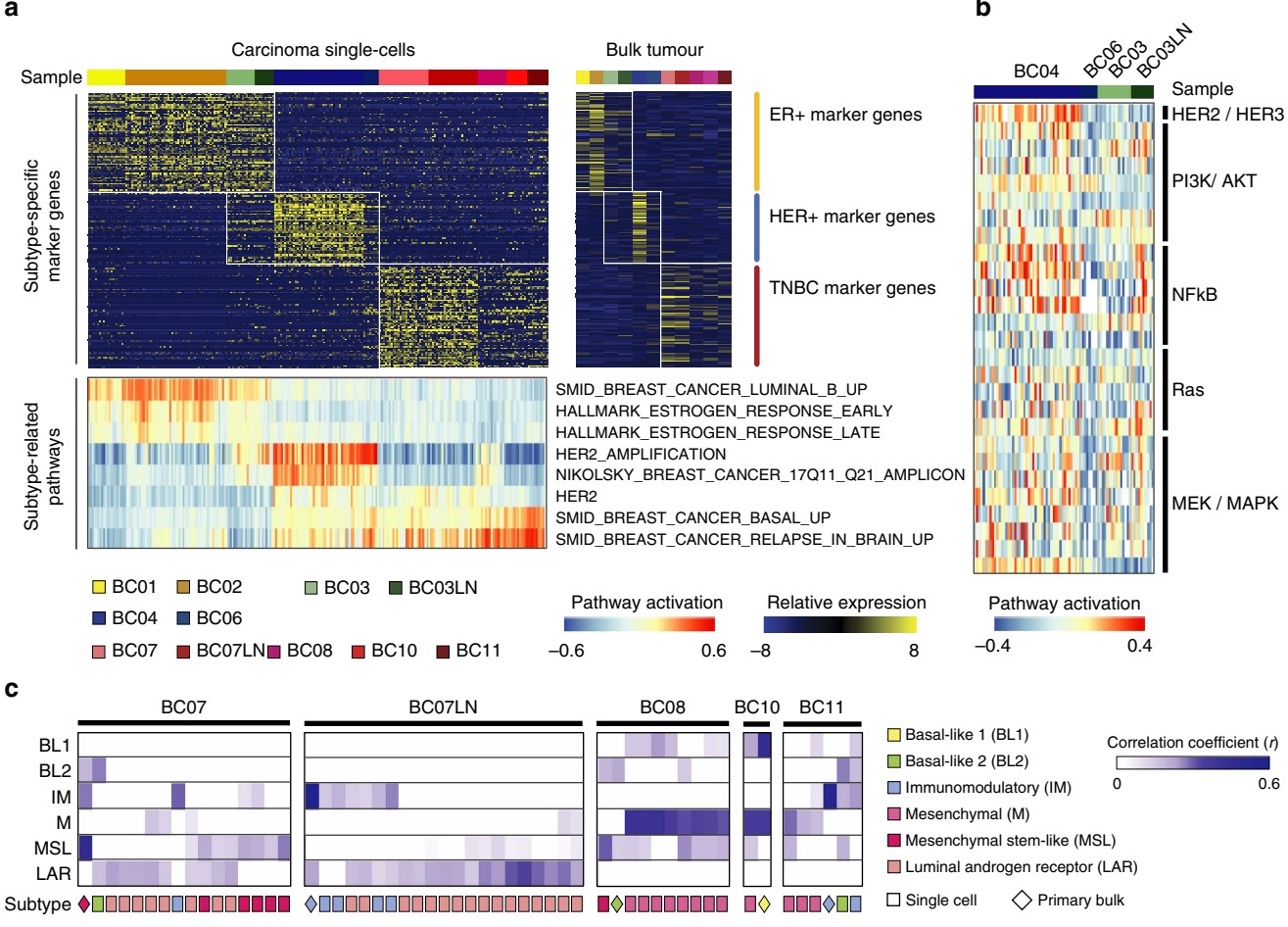

**Figure 5 | Subtype-specific gene expression profiling at single-cell resolution.** (**a**) LRT based on zero-inflated data and heatmap analysis (upper panels) identifying differentially expressed genes among different subtype tumours at a single-cell level (left) or in bulk (right). GSVA analysis with subtype-related pathways also indicates differential activation of pathways at a single-cell level (lower panel). (**b**) HER2/HER3 downstream signalling pathways (PI3K/AKT, NF-kB and RAS/MEK/MAPK) are highly activated in BC04 HER2+ tumour cells. Expression of PI3K and NF-kB pathway genes was upregulated in the lymph node metastasis for BC03 ER+HER2+ tumour cells. Pathway activation was determined by the GSVA enrichment score. (**c**) TNBCtype analysis[59] was applied only to TNBC tumours (BC07–BC11), which characterized individual cells as one or more of six different subtypes. Mixed subtype composition within a tumour indicates intratumoral heterogeneity comparable to intratumoral heterogeneity.

on zero-inflated data to extract differentially expressed genes for the luminal, HER2 or TNBC subtype carcinoma groups (Fig. 5a, top, and Supplementary Table 2a). Most carcinoma cells in all luminal tumours (BC01, BC02 and BC03) expressed high levels of ER and the canonical ER pathway genes[20,33,34] (Fig. 5a, bottom, and Supplementary Table 2b).

Carcinoma cells from the HER2+ (HER2; BC04 and BC06) and ER+HER2+ (luminal B; BC03) tumours expressed high levels of the *ERBB2/HER2* gene and genes located in the HER2 amplification region on chromosome 17q11-25. These carcinoma cells, however, had variable expression of HER2 signalling pathway genes[20]. Gene set variation analysis indicated higher expression of PI3K, NF-kB and MEK pathway genes for the BC04 carcinoma cells compared to others (Fig. 5b). The expression of PI3K and NF-kB pathway genes was low in the BC03 ER+HER2+ carcinoma cells, which was highly upregulated following lymph node metastasis.

Carcinoma cells from the TNBC tumour groups (BC07–11) exhibited variable upregulation of genes in basal pathways (Fig. 5a). Triple-negative breast cancer is known to be extremely heterogeneous in molecular, pathologic and clinical parameters. Although the results of initial subtype studies suggest that the majority of TNBC tumours belong to the basal-like subgroup,

TNBC and basal-like breast cancer may not represent identical tumour entities[35]. TNBC tumours can even be further classified into six different subgroups (basal-like 1, basal-like 2, immunomodulatory, luminal androgen receptor, mesenchymal and mesenchymal stem-like)[36]. On the basis of this TNBCtype classification scheme, TNBC carcinoma cells within a patient were assigned to multiple subgroups, thus showing extensive intratumoral heterogeneity (Fig. 5c). Interestingly, the TNBCtype distribution in the BC07 tumour changed on lymph node metastasis, suggesting a transition or selection of molecular signatures in different tumour microenvironments.

**Heterogeneity in tumour-infiltrating immune cells.** Most non-carcinoma cells were identified as immune cells based on their gene expression signatures (Fig. 2c,d). We further classified these 175 immune cells into three groups (Fig. 6a) by non-negative factorization clustering with immune cell type-specific gene sets[37] (Supplementary Figs 6, 7a and Supplementary Table 3). The largest group expressed immunoglobulins and B-cell-specific transcriptional factors, and many came from the tumour-infiltrating lymph nodes (cluster 1/B cells; Fig. 6a and Supplementary Table 4a). In the detailed analysis, two subclasses

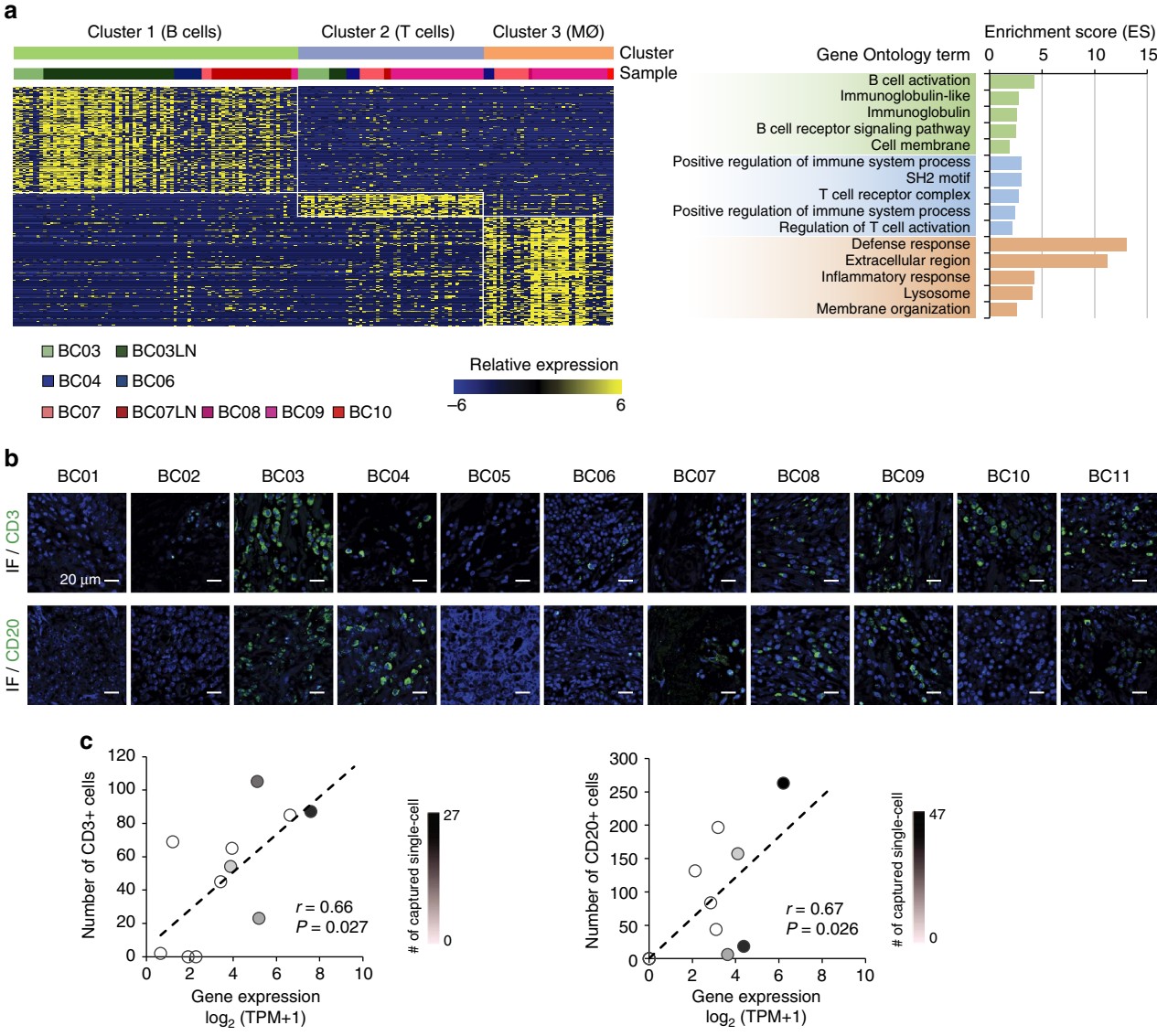

**Figure 6 | Identification of immune cell populations in the tumour microenvironment.** (**a**) Immune cell clusters were characterized by gene ontology terms. The cluster-specific genes extracted by LRT test (left) were associated with B cells, T cells or macrophages (MØ), respectively (right). (**b**) Immunofluorescence staining (IF) for CD3 or CD20 showing the infiltration of T or B lymphocytes in tumour tissues. Scale bar, 20 μm. (**c**) Immunofluorescence staining results show significant correlations with gene expression in bulk tumour samples (Pearson's r, 0.66 for CD3 and 0.67 for CD20, $P < 0.05$). The linear regression result is drawn as a dashed line. Number of captured single cells is marked with a colour key.

of B lymphocytes were identified, one with an expression signature of centroblasts/centrocytes[38] and the other with that of naive B lymphocytes (Supplementary Fig. 7b). The second group expressed T-cell receptors and T-cell-specific markers, most of which were captured from primary tumour tissues (cluster 2/T cells; Fig. 6a and Supplementary Table 4b). The third group also came from the primary tumour tissues and expressed markers for tissue macrophages (cluster 3/macrophages; Fig. 6a and Supplementary Table 4c).

The presence of tumour-infiltrating T and B lymphocytes was also assessed by immunostaining in tumour tissues with anti-CD3 (d + g + e) or anti-CD20 antibodies (Fig. 6b), which showed significant correlations with gene expression in the bulk tumour samples ($P < 0.05$; Fig. 6c). Among 10 tumour tissues with T-lymphocyte-specific gene expression and immunostaining, individual T-lymphocyte capture was successful in four tissue isolates. Individual B-lymphocyte capture was successful in four out of seven tissues. The results of additional marker staining for

T and B lymphocytes were consistent with the single-cell RNA-seq results (Supplementary Fig. 8). These data support the validity of gene expression profiling for cell type specification, but also implicate limitations in single-cell isolation from breast cancer tissues.

In the tumour-infiltrating T cells, we analysed gene expression signatures for T-cell activation and functional status (Fig. 7). T lymphocytes in four patient tumours manifested distinct patterns for naive, costimulatory, regulatory, exhaustion and cytotoxicity expression signatures[39–41]. The luminal B-type tumour (BC03) had T lymphocytes with naive/early costimulatory signatures in the primary tumour sites and T lymphocytes with more costimulatory molecule expression in the lymph nodes. One HER2 + tumour and two TNBC tumours (BC04 and BC07) were populated by T lymphocytes with the expression of regulatory T-cell markers, including IL2RA (also known as CD25)[41]. The third TNBC tumour (BC09) had two types of T lymphocytes, one with a predominant exhaustion

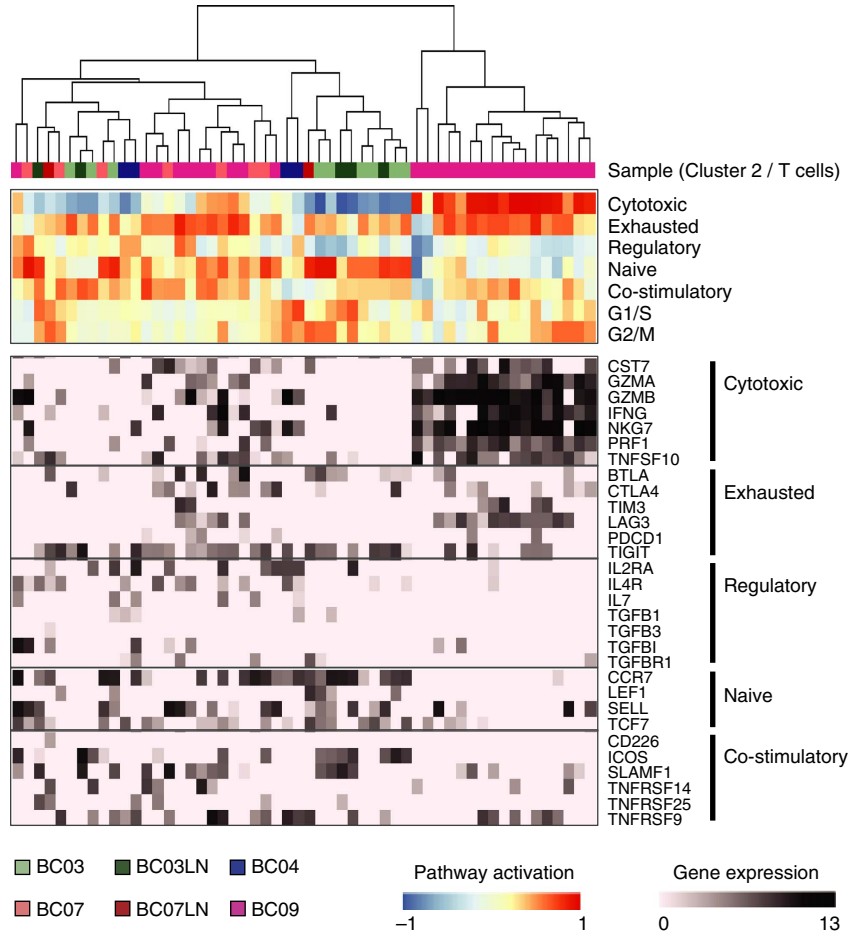

**Figure 7 | T-cell signatures in the tumour microenvironment.** Hierarchical clustering was performed using GSVA enrichment scores for gene sets for naive T cells, T-cell costimulation, regulatory cytokines and receptors, T-cell exhaustion and cytotoxicity (upper panel). Specific genes from used gene sets are presented in the lower panel.

signature and another exhibiting both exhaustion and cytotoxicity signatures. T cells with a high exhaustion signature are targets of immune checkpoint blockade in clinical oncology. In particular, blocking antibodies for PD-1 (*PDCD1*) or its ligand PD-L1 (*CD274*) were shown to have clinical efficacy in the treatment of melanoma and lung cancer[42]. However, PD-1 expression was modest in our data set, but expression of alternative inhibitory receptors such as TIGIT and LAG3 (ref. 43) was frequently detected.

The third immune cell group expressed transcripts for the monocyte/macrophage markers *CD14* and *CD68* (ref. 37), as well as phagocytic enzymes associated with macrophage function, suggestive of TAMs[44]. We identified genes that were differentially expressed in putative TAMs compared with the other immune cells from the single-cell transcriptome data (Supplementary Fig. 7a and Supplementary Table 4c). The selected genes were significantly enriched for genes involved in inflammation and defence mechanisms (Fig. 6a). Previous reports suggest that TAMs may show an immunosuppressive M2 signature, which promotes tumorigenesis by suppressing immune surveillance and inducing angiogenesis, rather than the activating M1-type signature[45]. Indeed, the putative TAM populations expressed many M2-type genes (Supplementary Fig. 7a) such as *CD163*, *MS4A6A* and *TGFBI* (ref. 45), in addition to genes known to promote tumour progression and angiogenesis such as *PLAUR*[13] and *IL-8* (ref. 46). Collectively, both innate and adaptive immune cell populations in the breast

cancer samples displayed immune-suppressive gene expression characteristics.

## Discussion

Using transcriptomic analysis of single-cell isolates from breast cancer, we could clearly separate the signatures of carcinoma and tumour-infiltrating immune cells, which together defined the characteristics of breast cancer. In the tumour cell analysis, we identified both heterogeneity and core gene expression signatures for subtype-specific breast cancer cells. We also classified non-tumour cells into three immune cell types with activating and suppressive gene expression signatures, suggesting dynamic immune cell interactions and a distinct immune system status in each tumour.

The role of the immune system in tumour progression has been extensively studied, and the results of these studies have provided the basis for successful immunotherapy in multiple cancer types[12,42]. Tumours are thought to evade natural immune surveillance either by immunologic ignorance or by active suppression. T-cell infiltration may be a key determinant by which the evasion pathway operates. Among immune cell infiltrates mostly obtained from TNBC tumours, we found a number of T cells with high cytokine and chemokine expression, which is indicative of ongoing immune responses. Interestingly, these T cells also manifested immuno-suppressive phenotypes of exhausted[39,40] or regulatory T cells[41]. Therapeutic

strategies to overcome T-cell exhaustion, that is, immune checkpoint blockade, have been developed that target CTLA-4 and PD-1/PD-L1, and these strategies have demonstrated significant efficacy in treating melanoma and non-small-cell lung cancer[42]. Clinical trials of these treatment strategies for other cancer types, including breast cancer, are in progress as a mono- or combination therapy. Search for new immune checkpoint targets is ongoing and early clinical trials for additional checkpoint molecules such as LAG3 (ref. 47) have begun. In our single-cell data set, most T cells expressed LAG3 and/or TIGIT[48] (Fig. 7), suggesting that they are potential targets for checkpoint inhibition.

Conventionally, the study of tumour-infiltrating immune cells has been conducted using predefined immune populations based on known surface marker expression. Here we used an inferred genomic feature, that is, RNA-seq-inferred CNV to separate carcinoma and immune cells without prior marker selection. This marker-free single-cell approach allows the use of gene expression signatures to define the carcinoma and immune cell populations, and makes it possible to demonstrate marker-unrestricted profiles and heterogeneity in the populations. There are potential errors in this approach, however, caused by a doublet formation between tumour and immune cells during single-cell capture and by a misclassification due to incomplete CNV inference. Using single-nucleotide polymorphism (SNP) information may reinforce the expression-based CNV inference by revealing additional genomic features of cancer cells such as loss of heterozygosity. Despite the potential errors, the marker-unrestricted capture and characterization of diverse tumour-associated populations may reveal new target populations for therapeutic intervention. Therefore, a sufficient level of cell capture without isolation bias would maximize the utility of marker-free cell identification. In our study, we collected only a small number of tentative cancer-associated fibroblasts or epithelial cells, and no endothelial cells probably due to the limitations of the cell isolation and capture methods. The partial representation of bulk tumour transcriptomes by those of single cells (Supplementary Fig. 2d,e) also suggests limitations of sampling in the current approach. To overcome this limitation and to profile the entire tumour microenvironment, cell isolation techniques enabling large scale, unbiased sampling need to be explored[32,49].

Many previously defined bulk tumour signatures were recapitulated at the single-cell level, especially for luminal and HER2-enriched subtype tumours, indicating that conventional transcriptome profiling in mixed bulk populations can successfully depict the intrinsic tumour properties. Nonetheless, single-cell profiling revealed individual tumour cell characteristics that might have been obscured in bulk analysis. First, we found that only one of the three tumours with *ERBB2/HER2* amplification had prominent activation of the HER2 downstream signalling pathway. The other two tumours were non-standard HER2 tumours, one had concurrent ER positivity and the other had been subjected to neoadjuvant chemo-/HER2 target therapy. Carcinoma cells from the ER and HER2 double-positive tumour showed both ER and HER2 expression with predominant ER downstream signalling pathway activation, classifying them as the luminal B type. Thus, these carcinoma cells may respond well to hormonal therapy at the sampling time point, yet may retain the potential to activate HER2 signalling and become resistant to hormonal therapy. Indeed, the ER + HER + carcinoma cells from the lymph node metastasis had higher gene expression for HER2 signalling pathways. The HER2-positive cells from the patient who had undergone neoadjuvant therapy showed low levels of HER signalling pathway activation but higher basal gene expression, resembling a TNBC tumour. These data suggest that single-cell gene expression profiling allows for a comprehensive understanding of the tumour cells and thus may help to develop effective treatment strategies.

Second, we identified gene expression characteristics of TNBC tumour cells, which have so far been elusive because of intrinsic tumour heterogeneity and the large number of infiltrating immune cells. Using single-cell RNA-seq, we confirmed the high level of transcriptome heterogeneity in TNBC tumour cells (Fig. 3b), which included signatures for basal, immunomodulatory, luminal androgen receptor, mesenchymal or stem-like gene expression (Fig. 5c). Notably, rare cell types with strong EMT and stemness signatures were identified within TNBC tumours (Fig. 4b), which may lead to tumour progression and metastasis. We also confirmed the high level of immune cell infiltration in TNBC tumours by histology, immunostaining and gene expression profiling of bulk tumours, and at a single-cell level. Considering the diverse immune cell types, single-cell expression profiling is particularly important for the accurate characterization of the tumour-infiltrating immune cells.

Altogether, these results demonstrate the scope and potential impact of intratumoral heterogeneity and suggest that single-cell transcriptome profiling can identify and characterize clinically important subpopulations to develop successful targeted treatments. Further, the results of our study indicate the need for large-scale single-cell gene expression profiling projects for the comprehensive characterization of heterogeneous tumours such as triple-negative breast cancer.

## Methods

**Patients and tumour specimens.** This study was approved by the Institutional Review Board of Seoul National University Hospital and Samsung Medical Center, and all patients provided signed informed consent for collection of specimens and detailed analyses of the derived genetic materials (Institutional Review Board no. 1207-119-420 and 2015-12-094-003). A total of 10 of 11 patients diagnosed with invasive ductal carcinoma underwent breast-conserving surgery or total mastectomy without prior treatment. Patient BC05 received neoadjuvant chemotherapy and Herceptin, and was thus excluded from subtype-specific analyses. In all, 11 primary tumour specimens (BC01–BC11) and 2 metastatic lymph nodes (BC03LN and BC07LN) were collected and processed for single-cell RNA sequencing. For BC09, two runs of single-cell RNA sequencing were performed and combined for downstream analysis. Molecular subtypes of tumours were predicted[50] using the R package genefu.

**Isolation of single cells and cDNA amplification.** Single-cell suspensions of breast cancer tissues or lymph node metastases were obtained by mechanical dissociation and enzymatic digestion on the day of the surgery. Dead cells were removed by Ficoll-Paque PLUS (17-1440-02, GE Healthcare, Uppsala, Sweden) separation, and 50,000 cells were loaded onto an individual 10–17 μm integrated fluidic circuit mRNA sequencing chip in the C1 Single-Cell Auto Prep System (100-5760, Fluidigm, San Francisco, CA, USA). Loaded chips were microscopically examined to verify single-cell loading. For cell lysis, cDNA synthesis and amplification, the SMARTer Ultra Low RNA Kit (634936, Clontech, Mountain View, CA, USA) was used following the manufacturer's instructions. RNA spike-ins 1, 4 and 7 from ArrayControl RNA Spikes (AM1780, ThermoFisher, Waltham, MA, USA) were added to the lysis mix. Amplified cDNAs were quantified and qualified using a Qubit 2.0 Fluorometer (Life Technologies, Carlsbad, CA, USA) and 2100 Bioanalyzer (Agilent Technologies, Santa Clara, CA, USA). In total, 579 single-cell cDNAs were subjected to RNA sequencing. Bulk RNAs were extracted from pooled cells ($\sim 1 \times 10^5$ tumour tissue isolates) or tumour tissues using the RNeasy Plus Micro kit (74034, Qiagen, Hilden, Germany), and 10 ng of total RNA was amplified with the SMARTer Ultra Low RNA Kit under the same conditions as used for single cells.

**Whole-exome sequencing and data processing.** Exomes were captured using the SureSelect XT Human All Exon V5 kit (5190-6208, Agilent). Sequencing libraries were constructed for the HiSeq 2500 system (Illumina) and sequenced using the 100-bp paired-end mode of the TruSeq Rapid PE Cluster kit and TruSeq Rapid SBS kit (PE-402-4001, Illumina). Exome-sequencing reads were aligned to the hg19 reference genome using BWA-0.7.10 (ref. 51). Putative duplications were marked by Picard-1.93 (http://sourceforge.net/projects/picard/files/picard-tools/1.118/). Sites potentially harbouring small insertions or deletions were realigned and recalibrated by applying the modules of GATK-3.2 (ref. 52) with known variant sites identified from phase I of the 1000 Genomes Project (http://www.1000genomes.org/) and dbSNP-137 (http://www.ncbi.nlm.nih.gov/

SNP/). We used MuTect-1.1.5 (ref. 53) to detect single nucleotide variations (SNVs) and the Control-FREEC software package[54] to detect CNVs. The whole exome sequencing (WES) coverage was $100\times$ for tumours and $50\times$ for paired blood samples.

**RNA sequencing and bioinformatics analysis.** Sequencing libraries were constructed using amplified cDNAs with the Nextera XT DNA Sample Prep Kit (FC-131-1024, Illumina) and sequenced using the HiSeq2500 system in 100-bp paired-end mode of the TruSeq Rapid PE Cluster kit and the TruSeq Rapid SBS kit. To assess the expression values of array control RNA spike-ins, reference sequences and the corresponding annotations were generated by merging three control RNA spike-ins (ThermoFisher) with the human genome reference sequences (hg19) and the GENCODE 19 annotations. The RNA reads were then aligned to the reference sequences using the 2-pass mode of STAR_2.4.0b (default parameters)[55], and relative gene expression was quantified as transcript per million (TPM) using RSEM v1.2.17 (default parameters)[56]. Isoform expression levels for each gene were summed to derive the TPM values. Quality control assessment of aligned single-cell RNA-seq reads was performed using RNA-SeQC[57], and the results are summarized in Supplementary Data 3. To remove cells with low-quality sequencing values, four filtering criteria were applied: (1) number of total reads; (2) mapping rate; (3) number of detected genes; and (4) portion of intergenic region. To remove genes with low expression values, the following steps were applied. First, TPM values $<1$ were considered unreliable and substituted with zero. Second, TPM values were $\log_2$-transformed after adding a value of one. Third, genes expressed in $<10\%$ of all tumour groups were removed. In total, 515 single cells and 17,779 genes passed the QC criteria. The filtered 17,779 genes were also used for bulk tumour analyses. As an estimate of the sensitivity and reproducibility of single-cell RNA sequencing, we obtained consistent $\log_2(\text{TPM}+1)$ ratios for RNA spike-in 1 (12,200 transcripts) and RNA spike-in 4 (912 transcripts; Supplementary Fig. 2a). RNA spike-in 7 with an estimated 62 transcript input was not detected. For transcriptome analysis, expression data were mean centred by subtracting the average $\log_2(\text{TPM}+1)$ value for each gene with the following exceptions: for RNA spike-in analysis; comparisons between RNA-seq and quantitative PCR (qPCR) results; detection of chromosomal expression patterns; measurement of intratumoral correlations; comparisons with immunofluorescence staining results; comparisons of immune marker expression; and application of self-normalizing tools such as Gene Set Variation Analysis (GSVA) and TNBCtype.

**Copy number inference from RNA-seq data.** To identify the distinct chromosomal gene expression pattern of cancer cells in comparison to putative non-carcinoma cells, expression profiles of normal breast tissues from GTEx portal (http://www.gtexportal.org/) were first transformed into $\log_2(\text{TPM}+1)$ values comparable to our data set. Second, the average gene expressions with their variations for all normal breast tissues were calculated for the genes detected in our filtered single-cell data. Third, the $Z$-scores of each gene were calculated by normalizing our single-cell data to the averaged expression profile of normal breast tissues. All genes were sorted by their chromosome number and start position. The chromosomal expression patterns were estimated from the moving averages of 150 genes as the window size and adjusted as centred values across genes.

**Correlations between genomic CNVs and inferred CNVs.** Comparisons of CNVs were performed for somatic CNVs and large-scale segments ($>10,000$ bp). To compare the CNVs between WES and RNA-seq in parallel, both CNVs were binned into 10 Mb window sizes, and inferred CNVs were averaged across single cells. The correlation coefficients were calculated by Pearson's correlation analysis using the R function.

**Pathway analysis.** To assess gene expression signatures and pathway activation, a non-parametric and unsupervised scoring algorithm called GSVA software in the R package[58] with the RNA-seq mode was used. We included GTEx normal breast tissue samples in the GSVA to assess relative pathway activation levels. For all gene sets, over-representation analysis was performed using the hypergeometric test, and those with $P$ values $<0.05$ were utilized.

**Breast cancer subtype-specific gene expression.** We applied the R package 'Seurat'[32] to analyse single-cell data and used the LRT based on zero-inflated data and the receiver-operating characteristic test to identify subtype-specific markers (whose average expression was larger than twofold and classification power (area under the curve) was higher than 0.7 with LRT $P<0.05$; Supplementary Table 2a–c). To show compartmental pathway activation in each subtype or identify the characteristic genes for HER2-enriched tumour cells compared to luminal HER2$+$ tumour cells, GSVA enrichment scores were calculated for subtype-related pathways or HER2/HER3 downstream signalling pathways from MSigDB v5.0.

**Triple-negative breast cancer subtyping.** Triple negative breast cancer cells were classified into six subtypes using TNBCtype (http://cbc.mc.vanderbilt.edu/tnbc/index.php)[59] (Fig. 5c). TNBCtype has six centroids for each subtype, defined by

2,188 subtype-signature genes and 386 training samples. By comparing a candidate sample with six centroids, TNBCtype provides Spearman correlation coefficients and $P$ values for each subtype. Some cells had high correlation coefficients ($P<0.05$) with more than one subtype, and thus were classified as multiple subtypes. Before the application of TNBCtype, genes that were not expressed in any single TNBC cells were removed. The input files were then uploaded without centring to avoid false ER$+$ tumour cell calling as a result of zero $ESR1$ expression in most of the TNBC single cells.

**Immune cell type-specific gene expression profiling.** Three immune cell subgroups were identified by non-negative factorization clustering[60] from 175 non-tumour cells using 412 genes annotated in 11 non-overlapping immune cell types[37] (Supplementary Table 3). To characterize the three immune cell clusters, the receiver-operating characteristic test and LRT based on zero-inflated data were performed using Seurat. Then, genes with a fold change $>2$ and an area under the curve $>0.7$ were obtained as cell type-specific genes (LRT $P<0.05$; Supplementary Table 4a–c). Gene ontology terms significantly enriched in cell type-specific genes were annotated by DAVID 6.7 (https://david.ncifcrf.gov/) with a default option. To further characterize T or B cells by functional status, GSVA analysis was performed with selected gene sets from the literature (Supplementary Table 5a–c).

**Immunofluorescence staining.** Immunofluorescence staining was carried out to assess the presence of tumour-infiltrating T or B cells in tumour tissues. T lymphocytes were double-stained with anti-CD3 (1:200; MA5-12577, Thermo Fisher, Waltham, MA, USA) and anti-MARK3 (1:100; PA5-29328, Thermo Fisher) antibodies in the formalin-fixed paraffin-embedded (FFPE) slides. B lymphocytes were double-stained with anti-CD20 (1:200; MA5-13141, Thermo Fisher) and anti-PRPSAP2 (1:50; PA5-31237, Thermo Fisher) antibodies. Alexa488-labelled-anti-mouse and Alexa568-labelled-anti-rabbit antibodies (1:50; Invitrogen) were used for double immunofluorescence with 4,6-diamidino-2-phenylindolecounter-staining. The numbers of CD3$+$ or CD20$+$ cells were assessed as average counts in three $0.125\,\text{mm}^2$ areas with maximal positive staining.

**Validation of RNA-seq data by qPCR.** qPCR was performed with the DELTA-gene assay (PN100-3035, Fluidigm) using cDNAs from 6 bulk and 185 single-cell samples. Primer sequences were designed using D3 software (Fluidigm) and are listed in Supplementary Table 6. Before comparison of qPCR and RNA-seq data, Ct values of 999 ($=$ not detected) were replaced with 'NA'. Ct values were negatively converted and $-20$ was set as the threshold value. These data represent the $\log_2$ expression level for qPCR comparable to $\log_2(\text{TPM}+1)$ for RNA-seq. The inter-relations were assessed by Pearson's correlation, Spearman's rank order correlation and linear regression analysis.

**Data availability.** The RNA-seq (single-cell and bulk) and bulk WES data have been deposited in the NCBI Gene Expression Omnibus database under the accession code GSE75688, and the bulk WES data have been deposited in the NCBI Sequence Read Archive under the accession code SRP067248. The TCGA Breast Invasive Carcinoma data referenced during the study are available in a public repository from the cBioportal website (www.cbioportal.org).

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

## Acknowledgements

We thank Eun Yoon Cho and Kyoung-Mee Kim (Department of Pathology, Samsung Medical Center) for the pathologic analysis. The biospecimens from Samsung Medical Center were provided by SMC BioBank. All sequencing was performed in Samsung Genome Institute. The study was supported by the National Research Foundation of Korea (2012M3A9B2029132 to W.-Y.P., 2014R1A1A3050273 and 2016R1A5A1011974 to H.-O.L, and 2015R1A2A2A01008264 to W.H.).

## Author contributions

W.C., H.H.E., H.-O.L. performed single-cell analysis and wrote paper; K.-M.L. and S.K. isolated single cells; H.S.R. performed tumour-infiltrating lymphocyte scoring; K.-T.K. provided computational analytical support; H.-B.L., J.E.L., Y.H.P. and Z.K. supported data interpretation; W.H. and W.-Y.P. conceived and guided the project.

## Additional information

**Competing interests:** The authors declare no competing financial interests.

