## [Peer Review File · Nature Communications]

Reviewers' comments:

Reviewer #1 (Remarks to the Author):

Summary: In this manuscript, the authors characterized cellular and transcriptional heterogeneity of breast cancer using single-cell RNA sequencing. They profiled the transcriptomes of 246 single cells obtained from four breast cancer patients, where each patient is classified into one of the four known sub-types of breast cancer. All the single cells were divided into tumour and non-tumour cells based on their genomic copy number alterations inferred from single-cell RNA-sequencing data. They showed that each subtype of breast cancer has distinguishing features, including the proportion of micro-environmental non-tumour cells, cellular heterogeneity of tumour cells, and gene expression signatures.

The present manuscript makes a valuable contribution by revealing cellular heterogeneity and gene expression characteristics of four subtypes of breast cancer with a single-cell resolution. However, their findings cannot be generalizable since the number of samples for each subtype is one. The authors clearly mentioned this limitation in the result and discussion.

Major points:

1. The authors used the CNV profiles inferred from the single-cell RNA-seq data to separate tumour cells from non-tumour ones. They argue that the inferred CNV profiles recapitulate the genomic CNV profiles estimated from whole exome sequencing, but did not quantify the similarity between the two CNV profiles. More specifically, the average of the inferred CNV profiles of single cells should have a significantly higher correlation with their matched genomic CNV profiles than those with non-matched profiles estimated from exome sequencing data. I'm also wondering whether unsupervised hierarchical clustering based on single-cell expression data (Supplementary Figure 3) can identify the non-tumour cluster.
2. The authors argue that TNBC tumour cells have the high level of heterogeneity in gene expression since these cells are classified into six subtypes of TNBC. However, they did not show that the six-subtype composition of TNBC is more diverse than that of other breast cancer subtypes.

Minor points:

1. Supplementary Fig. 1C: ERBB2 amplification in BC02 is not shown even though it is mentioned in the section of "Genomic profiles of four breast cancer subtypes for single-cell analysis" (ERBB2 amplification and missense mutations in ERBB2, KRAS, and TP53 (BC02))
2. Supplementary Fig. 2A: It seems that there is a batch effect between BC01 and BC03. The sentence in the third paragraph at page 6 ("Detection of constant ratios of two spiked-in RNAs throughout all single-cell RNA-seq experiments demonstrated a minimal batch effect between experiments") should be toned down since we cannot accurately determine whether there is a batch effect in this experimental design.
3. Supplementary Fig. 2D: This heatmap should be redrawn by using three colours (e.g. red for high positive correlation, white for no correlation, and blue for high negative correlation). Why do the averages of single cells of BC03 and BC04 have low correlations with their matched bulk tumours?
4. Supplementary Fig. 2E: Why is the multiple regression analysis used to measure the correlations between the average of single cells and bulk tumours? It looks like the number of explanatory variables is one.
5. Fig 1A, Fig 3C: The percentage of variance explained of PC1 and PC2 should be specified.
6. Fig 2C: What does the "Normal Tissues" represent?
7. Fig 4C: There are no labels in the second and fourth rows.
8. Fig 5D: In the y-axis, what does "correlation coefficient" mean? Why does the baseline start with 0.6?

Reviewer #2 (Remarks to the Author):

Authors perform single cell RNA sequencing of 246 single cells from 4 primary breast cancers and two associated lymph nodes representing the 4 main subtypes of breast cancer. Number of patient samples is small, but technology is expensive and labour intensive.

Results show not surprisingly, ITH at the single cell level, which is not a surprise in BC and has been demonstrated in GBM (Patel et al) as well as in BC (Navins et al), though not for TIL .

One interesting thing is that the ITH is demonstrated by the intermixing of all the cells from all patients, however, once the tumor and non-tumor cells are separated, the tumor cells seem quite concordant by subtype, with less ITH, and the immune cells are more similar to each other, rather than at the patient level. The authors separate the tumor vs non-tumor in an unbiased manner- using CN abnormalities, with confirmation by PCA and gene expression patterns that have been predefined in the literature. Figure 3 - what would happen if you took out the lymph node related cells BC03LN and BC04LN? I note that there is no yellow and very few green non-tumor cells.

Heterogeneity is shown at the single cell level in TNBC and in the Luminal B (ER+/HER2+ subtype), supporting data from Navins et al, Patel et al and Polyak K et al using StarFish. Please confirm that this TNBC data (Figure 5) is with tumor cells only, as immune cells are obviously going to be classified as "immunomodulatory" as the signature was derived from patients with high TILs in their primary TNBC tumor.

The most interesting thing about this data is it shows the feasibility of this technological approach to highlight new findings at the single cell level of non-tumor cells. The use of the Immune Hallmark gene sets, which contain over 1000 gene expression signatures, probably does not add much insight, except to show that the single cell results are plausible. More of an effort could have been made to try to understand mechanisms of immune activation; suppression. For example, what about T cell checkpoints that can be therapeutically targeted like PD-1 and PDL1. Are these T cells exhausted? How exhausted?

What was the activity of the PD-1/PD-L1 axis? Did all T-cells express PD-1 receptors? Did all tumour cells express PD-L1 in cases with high infiltration? What about markers of T-cell activation like Tbet and Eomes? Did some T-cells exhibit altered metabolic states (glycolytic versus fatty acid metabolism)? How did immune cells in the tumour differ compared to those in lymph nodes (understanding that the numbers are small and the variance high)? This surely deserves a sub-figure.

Figure 6: the authors claim differences in immune cells between BC03 (blue) vs. BC04 (pink- TNBC) tumors- what is the actual number of immune cells in the tumor (according to Figure 3, there are more for the TNBC?) I suspect there is a difference-more more in the latter: another explanation for their findings in Figure 6 is that BC02 (HER2+/ER-) and 04 (TNBC) have managed to activate immunity and are exhausted whilst BC03 (and 01) have not. ER+ are certainly less immunogenic compared with TNBC and ER-/HER2+ breast cancers. I am not sure that their explanation is correct and I would tone this down as based on two patients. What is the level of immune infiltrate per BC shown in Figure 1 C on H&E slides? Could the authors quantitate this in a standard way- see Salgado R et al, Annals of Oncology 2014-higher levels of immune infiltrate in TNBC and HER2+ breast cancers is associated with better prognosis. Were germinal centers actually seen on H&E for BC03? There is no IHC data to validate the gene expression findings.

Minor:

Title is a bit deceiving as most of the emphasis is on tumor cells and not the TILs (really Figure 6 only). Title should have the word "tumor" in it.

What is the depth of the exome sequencing as the BCs with more immune infiltration probably need deeper sequencing.

Figures: Colors need to be clearer- colors depicting BC subtypes vs enrichment vs relative expression was confusing at times on the figures.

It is confusing for me as a clinician to see each BC subtypes referred to as number. I think this should be replaced by TNBC, HER2, ER+ etc.

Please check that BC02 is not the ER+/HER2+ Luminal B vs BC03 the HER2 (ER-/HER2+): most data currently suggests that the ER+ have much lower infiltration than ER-/HER2+- this is not what is suggested in Figure 3C.

There is not much detail on how ANOVA was applied and what assumptions were made, or the justification for this approach (which is not necessarily optimal for single cell RNASeq). The limitations of ANOVA should

be mentioned (ie the high technical noise of single cell RNASeq, the problems caused by the mean-variance relationship for RNASeq).

In figure 2C, suggest either the data points or the box plots should be shown, but having both makes the graph cluttered.

The graph in figure 2d is interesting in that some of the tumour cells display expression of non-tumour cell markers, although the clusters are well demarcated. Is it possible that some tumour cells were doubled up with a lymphocyte (and vice versa) during the microfluidic separation? Somatic mutations in the tumour could perhaps be used here to detect contamination.

In figure 4b, it is not clear what the second row of the heat map (under the ESR1 row and the ER score row) refers to.

Papers supporting TILs as a robust prognostic factor in breast cancer, (see references in Salgado article) are not cited.

Reviewer #3 (Remarks to the Author):

The manuscript by Chung et al presents an exciting and important study that describes transcriptional and copy-number alteration analyses at the single cell level from cells extracted from tumors of four breast cancer patients, each representing one of the major breast cancer subtypes. This approach provides unprecedented resolution to characterize the cellular heterogeneity within each tumor, including cancer cells and also non-cancer cells from the tumor's microenvironment. The study showed that the carcinoma cells had transcriptional profiles that were characteristic of the "bulk" tumor subtype. Furthermore, they were able to detect different immune cell populations among the non-tumor cells from each tumor, capturing the heterogeneity of the immune state of individual tumors.

Although there is unquestionable excitement with this trailblazing study, skepticism remains about the reliability and generalizability of the presented results. Perhaps the biggest drawback of the study is that only one tumor from each subtype was analyzed. Given the genomic and transcriptional heterogeneity of breast tumors, in particular triple-negative breast cancer (TNBC) tumors, it is highly questionable whether any of the observations in this study can be generalized. Furthermore, it is not at all clear that 20-50 cells from each cancer can adequately represent the cell-to-cell heterogeneity within a tumor. Supplementary Fig. 2e appears to suggest that this may not be the case as the correlation between single-cell and bulk tumor transcriptional levels, as measured by the adjusted coefficient of determination of the linear regression, is between 0.5 and 0.65. It would be necessary to discuss more clearly both in the discussion and in the abstract the limitations due to single-cell transcriptional measurement noise and to the restricted sampling of cancer-to-cancer heterogeneity in this study.

Additional specific points to be addressed:

1. P. 6, Results, end of first subsection: "... suggests better representation of the tumor population with increasing number of single cells." It is true that this is what is shown by Suppl. Fig. 2e. However, the figure also suggests (R -squared < 0.7) that the heterogeneity of the bulk tumor is not sampled adequately, either due to inadequate number of cells or to biased single-cell capture.
2. P. 6, bottom: "Indeed a fraction with low tumor scores ...". Please explain in the methods how the tumor scores were calculated. Also the in-figure legend of Fig. 1b shows "Tumor score (ESTIMATE)" without providing any further explanation in the legend text or methods.
3. P. 7, 1st paragraph: "The separation of tumor vs non-tumor cells from CNV analysis..." This should probably read "cancer vs non cancer-cells" instead. Also it is not clear from the text or the methods, how the CNV patterns were used to differentiate cancer from non-cancer cells? Please explain.
4. P. 7, 2nd paragraph: "Altogether, among 246 cells, we estimated the capture of 141 epithelial breast cancer cells, 97 tumor-associated cells, and 8 non-tumor stromal or epithelial cells." How exactly were the

different cell types identified? Please explain.

5. P. 7, bottom (Fig. 3b): This figure shows, in particular for the TNBC sample and LN, that even after removal of the non-tumor cells, the pairwise correlations in the transcriptional profiles of single cells are low (< 0.6), suggesting greater cell-to-cell heterogeneity among cancer cells in these tumors. This is an observation worth mentioning in the text.

6. P. 8, top (and Suppl Fig 4): "... predicted the subtype of the TCGA breast cancer dataset with 91% accuracy ...". The label "POWER" in Table in Suppl. Fig 4c should be changed to "ACCURACY".

7. P. 8, top (Fig. 4b): It is not clear what the 2nd and 4th rows in Fig. 4b represent? Please add labels.

8. P. 8, 2nd paragraph (Fig. 4c). It is not clear from the univariable plots whether the "rare cells with a higher level of stemness, EMT, or angiogenesis..." are the same or different subpopulations. It would be informative to plot all the pairwise comparisons between the three scores for each of the 4 samples to determine co-occurrence or exclusivity of these features in single cancer cells.

9. P. 9, end 1st paragraph: "... our data emphasize the variable and dynamic nature of gene expression in TNBC tumor cells." Based on the lower part of Fig. 5a, it does not appear that TNBC cells have more variable transcriptional profiles than cells from the other two tumors. It is difficult to justify the above statement based on Fig. 5a.

10. Fig. 5d: it is not clear what this figure shows. The y-axis is some correlation coefficient, but it is not clear what that is. The legend does not provide any explanation either. Also the scale appears to be reversed. Is that really intentional?

11. Figure legends, Fig. 1: "Unsupervised principal component analysis (PCA) ..." Of what data? Is this based on the single-cell whole exome or on the transcriptome data? Please clarify in the legend.

Reviewers' comments:

Reviewer #1 (Remarks to the Author):

Summary: In this manuscript, the authors characterized cellular and transcriptional heterogeneity of breast cancer using single-cell RNA sequencing. They profiled the transcriptomes of 246 single cells obtained from four breast cancer patients, where each patient is classified into one of the four known sub-types of breast cancer. All the single cells were divided into tumor and non-tumor cells based on their genomic copy number alterations inferred from single-cell RNA-sequencing data. They showed that each subtype of breast cancer has distinguishing features, including the proportion of micro-environmental non-tumor cells, cellular heterogeneity of tumor cells, and gene expression signatures.

The present manuscript makes a valuable contribution by revealing cellular heterogeneity and gene expression characteristics of four subtypes of breast cancer with a single-cell resolution. However, their findings cannot be generalizable since the number of samples for each subtype is one. The authors clearly mentioned this limitation in the result and discussion.

Major points:

1. The authors used the CNV profiles inferred from the single-cell RNA-seq data to separate tumor cells from non-tumor ones. They argue that the inferred CNV profiles recapitulate the genomic CNV profiles estimated from whole exome sequencing, but did not **quantify the similarity between the two CNV profiles**. More specifically, the average of the inferred CNV profiles of single cells should have a significantly higher correlation with their matched genomic CNV profiles than those with non-matched profiles estimated from exome sequencing data. I'm also wondering whether **unsupervised hierarchical clustering based on single-cell expression data (Supplementary Figure 3) can identify the non-tumor cluster**.

Quantify the similarity between CNVs estimated from the single cell RNA-seq and bulk whole exome sequencing:

Following the reviewer's comments, we quantified the similarity between CNV profiles estimated from bulk exome and those inferred from single cell RNA-seq data. To compare the CNV estimation between WES and RNA-seq in parallel, both CNVs were binned into 10Mb window sizes, and inferred CNVs were averaged across single-cells. The correlation coefficients were calculated by Pearson's correlation analysis using the R function. The regions of significant CNVs for a representative tumor (BC01) (**Supplementary Fig. 3a**) and the correlation data for the CNVs from matched exome and averages of single cell RNA-seq (**Supplementary Fig. 3b**) are provided. In most patients except one, the inferred CNV has a higher correlation to the matched exome CNV than to non-matched CNVs. The exceptional case had very few CNV calls, which could be the reason for undistinguishable correlations to matched and non-matched exome CNVs.

Test whether unsupervised hierarchical clustering based on single-cell expression data can identify the non-tumor cluster:

As the reviewer pointed out, unsupervised hierarchical clustering largely distinguish the carcinoma and non-carcinoma cell clusters (**See Fig. 2b**). Thus unsupervised transcriptome clustering supports the result of CEP (chromosomal expression pattern)-based clustering. However, carcinoma cells undergoing epithelial-mesenchymal transition or surrounded by large number of immune infiltrates may show gene expression characteristics resembling stromal or immune cells. For those tumor cells, CEP would be an objective separation criteria, thus provide more robust means to distinguish the carcinoma from non-carcinoma cluster(s).

2. The authors argue that TNBC tumor cells have the high level of heterogeneity in gene expression since these cells are classified into six subtypes of TNBC. However, **they did not show that the six-subtype composition of TNBC is more diverse than that of other breast cancer subtypes.**

Intratumoral Heterogeneity in TNBC subtype tumors

The evidences of high level of tumor heterogeneity in TNBC compared to other subtypes are provided in two figures. In **Figure 1c**, large number of immune cell infiltration demonstrates tumor heterogeneity conferred by the microenvironment. In **Figure 3b**, low cell-to-cell correlations between carcinoma cells demonstrate the intrinsic intratumoral heterogeneity in TNBC.

By comparison, the six subtype composition in the revised **Figure 5c** does not show higher heterogeneity in TNBC compared to other subtypes. The main purpose of **Figure 5c** is to show extensive intratumoral heterogeneity within TNBC comparable to intertumoral (=inter-patient) heterogeneity. In this figure, we adopted a method developed by Lehmann and colleagues (JCI 2011, 121(7): 2750-2767, PMID21633166) which refined diverse molecular subtypes of triple negative breast cancer. The method is applicable only to TNBC type, as it excludes ER+ tumors from the analysis.

Minor points:

1. Supplementary Fig. 1C: **ERBB2 amplification in BC02** is not shown even though it is mentioned in the section of "Genomic profiles of four breast cancer subtypes for single-cell analysis" (ERBB2 amplification and missense mutations in ERBB2, KRAS, and TP53 (BC02))

-We corrected Supplementary Fig. 1c to include both amplification and missense mutation in ERBB2 for BC02 (**revised sample number BC06**). The figure is revised to include 11 patients as **Supplementary Fig. 1**.

2. Supplementary Fig. 2A: It seems that there is a batch effect between BC01 and BC03. The sentence in the third paragraph at page 6 ("Detection of constant ratios of two spiked-in RNAs throughout all single-cell RNA-seq experiments demonstrated a minimal batch effect between experiments") should be toned down since we cannot accurately determine whether there is a batch effect in this experimental design.

-As the reviewer pointed out, batch effect cannot be accurately determined in the current experimental design due to the complete confounding biology. We used TPM (transcripts per million) values for expression quantification and avoided additional normalization for most downstream analyses. To assure the quality of single cell RNA sequencing, we demonstrated the RNA spike-in expression for all the single cells and emphasized the constant ratios of the two spiked-in sequences. Following the reviewer's suggestion, we toned down the description to "Detection of constant ratios of two spiked-in RNAs assures the quality and consistency of all single-cell RNA-seq experiments" (Page 6, Line 23-25).

3. Supplementary Fig. 2D: This heatmap should be **redrawn by using three colours** (e.g. red for high positive correlation, white for no correlation, and blue for high negative correlation). **Why do the averages of single cells of BC03 and BC04 have low correlations with their matched bulk tumors?**

Redraw the heatmap using three color-scale

We redrew the heatmap in Supplementary Figure 2d following the reviewer's suggestion.

Why do the averages of single cells of BC03 and BC04 have low correlations with their matched bulk tumors?

The lower correlations in BC03 and BC04 (**revised sample index BC07 and BC07LN**) are likely due to the incomplete recovery of cellular components during the isolation process. In particular, we failed to obtain RNAs for the pooled cell isolates for BC04 (revised BC07), thus used tissues as bulk, unlike other samples. We commented on this in the **Supplementary Figure legend 2d**.

4. Supplementary Fig. 2E: **Why is the multiple regression analysis used** to measure the correlations between the average of single cells and bulk tumors? It looks like the number of explanatory variables is one.

-Supplementary Figure 2e demonstrates the increased representation of bulk population data with increasing number of single cells. This was achieved by random down-sampling of different number of single cell data, which required multiple regression analysis.

5. Fig 1A, Fig 3C: The percentage of variance explained of PC1 and PC2 should be specified.

-We specified the percentage of variance explained by PC1, PC2, and PC3 (added during the revision) in the revised Fig.1a and Fig.3c.

6. Fig 2C: What does the "Normal Tissues" represent?

-The "Normal Tissues" represent the breast tissue data from Genotype-Tissue Expression Project (GTEx, <http://www.gtexportal.org/home/>). They are generated from 183 mammary tissues. Due to the scarcity of single cell references, we used bulk tissue data as a normal reference. We added the information in the Figure legend 2c.

7. Fig 4C: There are no labels in the second and fourth rows.

-It is likely that the reviewer meant Fig. 4b and the label was at the bottom side. As the reviewer pointed out, the labels for Fig. 4b in the original submission were confusing. In the revised manuscript, we incorporated Fig. 4a and 4b, and simplified figures to have clear labels.

8. Fig 5D: In the y-axis, what does "correlation coefficient" mean? Why does the baseline start with 0.6?

Explain correlation coefficient in Fig. 5d

We classified TNBC single cells into 6 subtypes using 'TNBCtype' tool (PMID22872785, <http://cbc.mc.vanderbilt.edu/tnbc/index.php>) with slight modification. The TNBCtype tool generates six centroids for each subtype, defined by 2,188 subtype-signature genes and 386 training samples. By comparing candidate samples with six centroids, TNBCtype specifies a subtype with the highest **Spearman correlation to the centroid**. We added the method description in the revised manuscript.

As stated in the response to the second major point, the main purpose of Fig. 5d (revised Fig. 5c) is to show extensive intratumoral heterogeneity within TNBC comparable to intertumoral (=inter-patient) heterogeneity. For clear representation, we replaced the bar graph with a heatmap to show mixed and multiple TNBC subtype distribution within each patient tumor.

Reviewer #2 (Remarks to the Author):

Authors perform single cell RNA sequencing of 246 single cells from 4 primary breast cancers and two associated lymph nodes representing the 4 main subtypes of breast cancer. Number of patient samples is small, but technology is expensive and labour intensive.

Results show not surprisingly, ITH at the single cell level, which is not a surprise in BC and has been demonstrated in GBM (Patel et al) as well as in BC (Navins et al), though not for TIL .

One interesting thing is that the ITH is demonstrated by the intermixing of all the cells from all patients, however, once the tumor and non-tumor cells are separated, the tumor cells seem

quite concordant by subtype, with less ITH, and the immune cells are more similar to each other, rather than at the patient level. The authors separate the tumor vs non-tumor in an unbiased manner- using CN abnormalities, with confirmation by PCA and gene expression patterns that have been predefined in the literature. **Figure 3 - what would happen if you took out the lymph node related cells BC03LN and BC04LN?** I note that there is no yellow and very few green non-tumor cells.

PCA after removal of the lymph node data:

In the revised manuscript, we sequenced more patient samples, which increased the number of both tumor and immune cells. As the reviewer pointed out, there are large number of immune cells in the lymph nodes and taking out the lymph node data resulted in the decrease of immune cell populations.

Thus, **removal of lymph node cells changed the PCA plot for immune cells but had a negligible effect on the PCA for tumor cells. We provide the revised Fig. 3c and an equivalent figure without lymph node cells for reviewers only.**

[For Reviewers only. Principal component analysis before and after the removal of lymph node cells. In both analyses, patient-specific tumor cell clustering and mixed immune cell clustering can be demonstrated.]

Heterogeneity is shown at the single cell level in TNBC and in the Luminal B (ER+/HER2+ subtype), supporting data from Navins et al, Patel et al and Polyak K et al using StarFish. Please **confirm that this TNBC data (Figure 5) is with tumor cells only**, as immune cells are obviously going to be classified as "immunomodulatory" as the signature was derived from patients with high TILs in their primary TNBC tumor.

Confirm that this TNBC data (Figure 5) is with tumor cells only:

We agree with the reviewer that immune cells could be mistakenly categorized as tumor cells with "immunomodulatory" phenotype. To avoid the mis-interpretation we used inferred genomic features of tumor cells. The chromosomal gene expression pattern in Fig. 2a supports those cells are tumor with the patient specific CNV (RNA-inferred CNV). Additionally, somatic mutations such as single nucleotide variation (SNV) detected in the RNA-seq could be used to

mark tumor cells. Fifteen out of 18 immunomodulatory tumor cells had somatic mutations in the coding sequences (ranging 1–8 per cell). However, we only used inferred CNV for tumor cell selection because SNV detection in single cell RNA-seq was unreliable due to allele drop-out events and sequencing errors.

The most interesting thing about this data is it shows the feasibility of this technological approach to highlight new findings at the single cell level of non-tumor cells. The use of the Immune Hallmark gene sets, which contain over 1000 gene expression signatures, probably does not add much insight, except to show that the single cell results are plausible. More of an effort could have been made to try to understand mechanisms of immune activation; suppression. For example, **what about T cell checkpoints that can be therapeutically targeted like PD-1 and PDL1. Are these T cells exhausted? How exhausted?**

What was the activity of the PD-1/PD-L1 axis? Did all T-cells express PD-1 receptors? Did all tumor cells express PD-L1 in cases with high infiltration? What about markers of T-cell activation like Tbet and Eomes? Did some T-cells exhibit altered metabolic states (glycolytic versus fatty acid metabolism)? How did immune cells in the tumor differ compared to those in lymph nodes (understanding that the numbers are small and the variance high)? This surely deserves a sub-figure.

Present data for T cell activation and exhaustion

As the reviewer pointed out, understanding the status of immune cells is an important issue for the therapeutic potential, and single cell sequencing provides unprecedented opportunities for tumor-infiltrating immune cell profiling. Thus, we followed the reviewer's suggestion and analyzed immune cells, especially T cells in detail.

First, we extended the previous Fig. 6b to determine the **cytotoxicity vs. exhaustion signatures** of tumor-infiltrating T cells in four out of eleven patients (**revised Fig. 6d**). We did not recover any T cells in the remaining seven patients (limitation in the single cell isolation). We found differences in the balance between T cell cytotoxicity and exhaustion, as recently demonstrated in the single cell melanoma study (PMID27124452, <http://science.sciencemag.org/content/352/6282/189>).

Second, we examined the **expression of immune checkpoint genes**, which make up the T cell exhaustion signature (Fig. 6d). In our dataset, T cells from only one patient had PD-1 (*PDCD1*) expression yet those from other patients expressed variable levels of other inhibitory receptors such as BTLA, CTLA4, LAG3, and TIGIT. Notably, TIGIT and LAG3 were expressed by the largest number of T cells, suggesting them as potential targets for immune therapy in breast cancer.

Check gene expression for T cell activation and metabolic state

As the reviewer pointed out, T cell activation and differentiation determine their functional status. Following the reviewer's suggestion, we analyzed expression signatures for T cell activation, cell cycle progression, and metabolic state. In the revised Figure 6d, gene expression signatures are presented for naïve T cells, regulatory cytokines and receptors, T cell costimulatory molecules, and cell cycle progression. **Gene expression for Tbet (gene symbol, TBX21) and EOMES remained low and detected in only few cells. Geneset expression representing the metabolic state (glycolytic vs.**

fatty acid metabolism in mSigDB, software.broadinstitute.org/gsea/msigdb/) remained relatively low and constant in our dataset (presented for Reviewers only).

[For Reviewers only. Gene set variation analysis of metabolism signatures and expression of T cell transcription factors, TBX21 and EOMES are drawn along with other T cell signatures.]

Compare T cells in the primary tumor sites vs. in the lymph nodes

In our dataset, we isolated tumor cells from lymph node metastasis from two patients, a luminal B (ER+HER2+) and a TNBC type. Tumor cells from the lymph node metastasis recapitulated the gene expression characteristics in the primary site for each tumor, and very few common gene expression signatures were found between the two lymph nodes. Thus, we compared gene expression for the primary site and lymph node in each tumor separately, and found higher HER2 pathway activation in the luminal B lymph node (**Fig. 5b**), and higher immunomodulatory/luminal androgen receptor signatures in the TNBC lymph node metastasis (**Fig. 5c**).

Figure 6: the authors claim differences in immune cells between BC03 (blue) vs. BC04 (pink-TNBC) tumors- what is the actual number of immune cells in the tumor (according to Figure 3, there are more for the TNBC?) I suspect there is a difference-more more in the latter: another explanation for their findings in Figure 6 is that BC02 (HER2+/ER-) and 04 (TNBC) have managed to activate immunity and are exhausted whilst BC03 (and 01) have not. ER+ are certainly less immunogenic compared with TNBC and ER-/HER2+ breast cancers. I am not sure that their explanation is correct and I would **tone this down** as based on two patients.

What is the level of immune infiltrate per BC shown in Figure 1

C on H&E slides? Could the authors quantitate this in a standard way- see Salgado R et al, Annals of Oncology 2014-higher levels of immune infiltrate in TNBC and HER2+ breast cancers is associated with better prognosis. **Were germinal centers actually seen on H&E for BC03? There is no IHC data to validate the gene expression findings.**

Discuss alternative explanations for T cell phenotypes

We agree with the reviewer that there are many possible explanations for our findings and the conclusion was presumptuous. Following the reviewer's suggestion, we toned down our initial interpretation and presented more data and discussion on the infiltrating immune cells. We split original Figure 6b to **Figure 6d for T cell phenotype** and **Supplementary Figure 7 for B cell phenotype** description.

Evaluate the level of immune cell infiltration

In response to the request of quantification of immune infiltrates in tumor tissues, we evaluated H&E slide for the percent of immune cell infiltration (**Fig 1c**). We also performed antibody staining for T cells with pan anti-CD3 (d,e, and g) antibodies and for B cells with a monoclonal anti-CD20 antibody (**revised Fig 6b**). The cell counts from the immunostaining, expression levels from bulk RNA sequencing, and the recovery of single cells are plotted as revised **Figure 6c**. Some discordance is noted between the estimated level of immune cell infiltration and tissue staining, which is likely ascribed to the limitations in sampling in both methods. In particular, the recovery of immune

cells in the single cell sequencing can be influenced by cell numbers/proportions and other factors such as the rigidity of tumor tissues.

Immunohistochemistry and germinal center detection

In response to the reviewer's request for the immunohistochemistry data, we performed immunofluorescence staining instead, to determine the co-staining pattern of cell lineage markers and other marker expression. Co-staining results of CD3 and MARK3 expression, and those of CD20 and PRPSAP2 are provided as **Supplementary Figure 8**. Lymph node specimens were not available as tissue sections, and the presence of germinal centers could not be determined. We toned down comments on germinal centers accordingly.

Minor:

Title is a bit deceiving as most of the emphasis is on tumor cells and not the TILs (really Figure 6 only).

Title should have the word "tumor" in it.

-We changed the title to **"Single-cell RNA-seq enables comprehensive tumor and immune cell profiling in primary breast cancer"**.

What is the depth of the exome sequencing as the BCs with more immune infiltration probably need deeper sequencing.

-The average depth of the tumor exome sequencing was 100X. All the sequencing parameters are provided in the materials and methods.

Figures: Colors need to be clearer- colors depicting BC subtypes vs enrichment vs relative expression was confusing at times on the figures.

-With increase in the number of patients, color coding for each patient sample could be even more confusing. To minimize the confusion, we kept consistency in the color usage throughout the figures and marked description over color coding when possible.

It is confusing for me as a clinician to see each BC subtypes referred to as number. I think this should be replaced by TNBC, HER2, ER+ etc.

-Following the reviewer's suggestion, we replaced most of the number description to the tumor types in the revised manuscript. At times we used both labels.

Please check that BC02 is not the ER+/HER2+ Luminal B vs BC03 the HER2 (ER-/HER2+): most data currently suggests that the ER+ have much lower infiltration than ER-/HER2+ - this is not what is suggested in Figure 3C.

-Please note that we renumbered the tumors in the revised manuscript. The original BC02 is BC04 in the revised manuscript. The original BC03 is still BC03. We checked the tumor types and confirmed the original description. As the reviewer pointed out, ER+ tumors are known to have fewer immune cell infiltration compared to ER- breast cancer. The ER+HER2+ luminal B tumor in our dataset had both ER expression and HER2 amplification. This particular tumor also had lymph node metastasis which may explain the large number of tumor-infiltrating T and B lymphocytes. It is likely that multiple factors –such as ER expression, HER2/ERBB2 expression, tumor stage, and lymph node metastasis- influence the extent of immune cell infiltration.

There is not much detail on how ANOVA was applied and what assumptions were made, or the justification for this approach (which is not necessarily optimal for single cell RNASeq). The limitations of ANOVA should be mentioned (ie the high technical noise of single cell RNASeq, the problems caused by the mean-variance relationship for RNASeq).

Why ANOVA?

As the reviewer pointed out, there are analytical tools optimized for the zero-inflated single cell RNA-seq data. In the revised Fig. 5a, we applied LRT test based on zero-inflated data ("bimod", default) from the R package Seurat (PMID25867923, Spatial reconstruction of single-cell gene expression data, Nat. Biotechnol. 2015 33(5):495-502) to retrieve differentially expressed genes between breast cancer subtypes at a single cell level.

In figure 2C, suggest either the data points or the box plots should be shown, but having both makes the graph cluttered.

-Following the reviewer's suggestion, we simplified them to box plots.

The graph in figure 2d is interesting in that some of the tumor cells display expression of non-tumor cell markers, although the clusters are well demarcated. Is it possible that some tumor cells were doubled up with a lymphocyte (and vice versa) during the microfluidic separation?

Somatic mutations in the tumor could perhaps be used here to detect contamination.

In figure 4b, it is not clear what the second row of the heat map (under the ESR1 row and the ER score row) refers to.

Use somatic mutations to check contamination

As the reviewer pointed out, doublet problem cannot be completely avoided in the single cell analysis. To minimize doublet contamination, we scanned enlarged capture images for all the single cells and excluded 36.7% cells (on average) as potential doublets. Following the reviewer's suggestion, we also assessed somatic mutations from the single cell RNA-seq. As shown in the Reviewers only figure, very few somatic mutation calls could be made (due to drop-out events) and thus could not be used to distinguish tumor from non-tumor cells. As the contamination is an important issue in the single cell analysis, we added discussion on this point in Page 11, Line 33-35.

"There are potential errors in this approach, however, caused by a doublet formation between tumor and immune cells during single cell capture and by a misclassification due to incomplete CNV inference."

[For Reviewers only. Somatic SNVs detected in the single cell RNA-seq]

Clarify figure labeling

In Figure 4b, ESR1 and ERBB2 refer to gene expression level for each gene, whereas ER and HER2 scores refer to geneset expression representing ER or HER2 pathway activation. In the revised manuscript we simplified and combined Figure 4a and 4b to avoid confusion in the labeling.

Papers supporting TILs as a robust prognostic factor in breast cancer, (see references in Salgado article) are not cited.

-We referred to several articles and reviews describing TILs, including

Salgado's. Reviewer #3 (Remarks to the Author):

The manuscript by Chung et al presents an exciting and important study that describes transcriptional and copy-number alteration analyses at the single cell level from cells extracted from tumors of four breast cancer patients, each representing one of the major breast cancer subtypes. This approach provides unprecedented resolution to characterize the cellular heterogeneity within each tumor, including cancer cells and also non-cancer cells from the tumor's microenvironment. The study showed that the carcinoma cells had transcriptional profiles that were characteristic of the "bulk" tumor subtype. Furthermore, they were able to detect different immune cell populations among the non-tumor cells from each tumor, capturing the heterogeneity of the immune state of individual tumors.

Although there is unquestionable excitement with this trailblazing study, skepticism remains about the reliability and generalizability of the presented results. Perhaps the biggest drawback of the study is that only one tumor from each subtype was analyzed. Given the genomic and transcriptional heterogeneity of breast tumors, in particular triple-negative breast cancer (TNBC) tumors, it is highly questionable whether any of the observations in this study can be generalized. Furthermore, it is not at all clear that 20-50 cells from each cancer can adequately represent the cell-to-cell heterogeneity within a tumor. Supplementary Fig. 2e appears to suggest that this may not be the case as the correlation between single-cell and bulk tumor transcriptional levels, as measured by the adjusted coefficient of determination of the linear regression, is between 0.5 and 0.65. It would be necessary to discuss more clearly both in the discussion and in the abstract the limitations due to single-cell transcriptional measurement noise and to the restricted sampling of cancer-to-cancer heterogeneity in this study.

-We are grateful for both appreciation and criticism on the study. The reviewer's points are well taken as important issues in the single cell sequencing studies. In the revised manuscript, we increased the number of patients to strengthen our findings. However, we acknowledge that it is far short for the generalization, and discussed limitations in the sampling in the discussion section. (Page 11, Line 38-46)

"..... Therefore, a sufficient level of cell capture without isolation bias would maximize the utility of marker-free cell identification. In our study, we collected only a small number of tentative cancer-associated fibroblasts or epithelial cells, and no endothelial cells probably due to the limitations of the cell isolation and capture methods. The partial representation of bulk tumor transcriptomes by those of single cells (**Supplementary Fig. 2d and e**) also suggests limitations of sampling in the current approach. To overcome this limitation and to profile the entire tumor microenvironment, cell isolation techniques enabling large scale, unbiased sampling need to be explored."

Additional specific points to be addressed:

1. P. 6, Results, end of first subsection: "... suggests better representation of the tumor population with increasing number of single cells." It is true that this is what is shown by Suppl. Fig. 2e. However, the figure also suggests (R -squared < 0.7) that the heterogeneity of the bulk

tumor is not sampled adequately, either due to inadequate number of cells or to biased single-cell capture.

-We agree with the reviewer on this point and changed the comment to acknowledge both increasing power and the limitation of incomplete sampling.

(Page 6, Line 27-34) "Comparisons between tumor tissue and tissue isolates (**Supplementary Fig. 2c**), or between the averages of single cells and corresponding pooled samples (**Supplementary Fig. 2d**) demonstrated partial but significant correlations. Multiple regression analyses of the transcriptomes of different sized pools of single cells to those of bulk tumors (**Supplementary Fig. 2e**) provided a better representation of the tumor population with an increasing number of single cells. Altogether, single-cell RNA-seq could illustrate a significant portion of the tumor entity, yet tumor components were lost during the single cell isolation or sequencing processes."

2. P. 6, bottom: "Indeed a fraction with low tumor scores ...". Please explain in the methods how the tumor scores were calculated. Also the in-figure legend of Fig. 1b shows "Tumor score (ESTIMATE)" without providing any further explanation in the legend text or methods.

Explain ESTIMATE

We added brief description of "tumor score from the ESTIMATE, Estimation of Stromal and Immune cells in Malignant Tumors genes (PMID24113773, <http://www.nature.com/ncomms/2013/131011/ncomms3612/full/ncomms3612.html>)" in the text and

Figure legend 2c. ESTIMATE algorithm is based on single sample Gene Set Enrichment Analysis

using representative gene expression for stromal and immune cells to infer normal cell fraction and tumor purity. We used the ESTIMATE score as tumor score. Stromal score was calculated from Gene set variation analysis with 141 stromal genes, and immune score with 141 immune genes. All the analyses were run on the single cell expression data along with GTEx normal breast tissue expression data.

(Page 7, Line 18-21) "To further delineate the identity of carcinoma and non-carcinoma cells, we analyzed the expression of tumor-associated stromal or immune gene sets proposed in the tumor purity estimation, ESTIMATE²⁴. Most of the non-carcinoma cells scored high for the immune signature (**Fig. 2c**)."

(**Figure legend 2c**) "Carcinoma cells identified in **a**, scored low for stromal and immune signatures whereas non-carcinoma cells scored high for immune signatures.

Tumor score was

inferred from the stromal and immune signature using ESTIMATE algorithm.²⁴ Normal tissues

represent 183 mammary tissue data from GTEx portal (<http://www.gtexportal.org/>)."

3. P. 7, 1st paragraph: "The separation of tumor vs non-tumor cells from CNV analysis..." This should probably read "cancer vs non cancer-cells" instead. Also it is not clear from the text or the methods, how the CNV patterns were used to differentiate cancer from non-cancer cells? Please explain.

Explain CNV inference

We changed the description to "cancer vs. non cancer-cells following the reviewer's suggestion. We added detailed description for the use of chromosomal expression pattern for the inference of copy number variation in the method. We used RNA-seq based CNV as an inferred genomic alteration in cancer cells. Both inferred CNV and SNV would mark cancer cells, and we used inferred CNV (Fig. 2a) as very few SNVs could be detected due to allele drop-outs.

4. P. 7, 2nd paragraph: "Altogether, among 246 cells, we estimated the capture of 141 epithelial breast cancer cells, 97 tumor-associated cells, and 8 non-tumor stromal or epithelial cells." How exactly were the different cell types identified? Please explain.

Explain cell type specification

In the revised manuscript, the number has been changed to 515 cells in total, 317 carcinoma cells and

198 non-carcinoma cells. Among 198 non-carcinoma cells, 175 cells were estimated to be immune cells, and 23 non-immune (stromal or epithelial) cells. We made cell type decision in two-stepwise

manner. In the first step, unsupervised hierarchical clustering was applied to separate clusters of carcinoma and the non-carcinoma cluster, based on the chromosomal expression pattern as inferred CNV. In the second step, we used 141 stromal and 141 immune genes

(PMID24113773,

<http://www.nature.com/ncomms/2013/131011/ncomms3612/full/ncomms3612.html>) to further

categorize non-carcinoma cells. In the Gene Set Variation Analysis (GSVA), most non-carcinoma cells had high immune GSVA score but low stromal score. Twenty three outlier cells with low immune score were classified as non-carcinoma, non-immune cells.

5. P. 7, bottom (Fig. 3b): This figure shows, in particular for the TNBC sample and LN, that even after removal of the non-tumor cells, the pairwise correlations in the transcriptional profiles of single cells are low (< 0.6), suggesting greater cell-to-cell heterogeneity among cancer cells in these tumors. This is an observation worth mentioning in the text.

-Thanks for bringing up the important point. Extended data from 11 patients also demonstrate low intratumoral correlations in TNBC single cells. We described the results in page 7, line 35-37.

"Carcinoma cells from TNBC type tumors (BC07-11) showed lower cell-to-cell correlations compared to those from luminal or HER2 types (**Fig. 3b**), suggesting a high level of intratumoral heterogeneity in TNBC."

6. P. 8, top (and Suppl Fig 4): "... predicted the subtype of the TCGA breast cancer dataset with 91% accuracy ...". The label "POWER" in Table in Suppl. Fig 4c should be changed to "ACCURACY".

-We changed the label to "Accuracy" in the **revised Supplementary Fig. 4**.

7. P. 8, top (Fig. 4b): It is not clear what the 2nd and 4th rows in Fig. 4b represent? Please add labels.

-They were gene expression levels for ER (*ESR1*) and HER2 (*ERBB2*). To avoid confusion, we simplified Figure 4b and incorporated into **revised Figure 4a**. Instead of using both gene expression and module scores (pathway activation), we used only module scores to show mixed subtypes in tumors.

8. P. 8, 2nd paragraph (Fig. 4c). It is not clear from the univariable plots whether the "rare cells with a higher level of stemness, EMT, or angiogenesis..." are the same or different subpopulations. It would be informative to plot all the pairwise comparisons between the three scores for each of the 4 samples to determine co-occurrence or exclusivity of these features in single cancer cells.

Provide pairwise comparisons of cancer signatures and mark aggressive cancer

cells As the reviewer pointed out, the presence of "highly metastatic cancer stem cells" would be an important issue, hence we put pairwise comparisons between the signatures as **revised Figure 4b**.

Positive correlations between stemness, EMT, and angiogenesis were detected. We labeled carcinoma cells with two of the aggressive signatures in top 5% as a “highly aggressive” carcinoma subpopulation. We moved the univariable plots to Supplementary Figure 5.

9. P. 9, end 1st paragraph: "... our data emphasize the variable and dynamic nature of gene expression in TNBC tumor cells." Based on the lower part of Fig. 5a, it does not appear that TNBC cells have more variable transcriptional profiles than cells from the other two tumors. It is difficult to justify the above statement based on Fig. 5a.

-We agree with the reviewer that Fig. 5a does not present more variable gene expression in TNBC tumor cells compared to other subtype breast cancer. The purpose of Fig. 5a was to find core signatures in each breast cancer subtype at cellular level. As the reviewer brought out in the specific point #5, TNBC tumor cells demonstrated higher level of intratumoral heterogeneity compared to other subtypes in overall gene expression (Fig. 3b).

10. Fig. 5d: it is not clear what this figure shows. The y-axis is some correlation coefficient, but it is not clear what that is. The legend does not provide any explanation either. Also the scale appears to be reversed. Is that really intentional?

-The purpose of the Fig. 5d (revised Fig. 5c) is to show extensive intratumoral heterogeneity within TNBC comparable to intertumoral (=inter-patient) heterogeneity using ‘TNBCtype’ tool (PMID22872785, <http://cbc.mc.vanderbilt.edu/tnbc/index.php>). The TNBCtype tool generates six centroids for each subtype, defined by 2,188 subtype-signature genes and 386 training samples. By comparing candidate samples with six centroids, TNBCtype specifies a subtype with the highest Spearman correlation to the centroid. For clear presentation, we replaced the bar graph with a heatmap showing correlation coefficients for multiple subtypes within each patient tumor. We added the method description in the revised manuscript.

11. Figure legends, Fig. 1: "Unsupervised principal component analysis (PCA) ..." Of what data? Is this based on the single-cell whole exome or on the transcriptome data? Please clarify in the legend.

-Unsupervised principal component analysis was performed on single cell transcriptome data from 11 patients in the revised manuscript. We corrected the figure legend accordingly.

Reviewers' comments:

Reviewer #1 (Remarks to the Author):

The revised manuscript addressed most of my concerns except some minor points.

Minor points:

1. Supplementary Fig. 2E: The description regarding the multiple regression analysis is still confusing. I think the expression levels of each single cell of a gene were used as explanatory variables to predict the expression level of bulk tumours. However, most readers would expect the regression analysis between the average of single cells and bulk tumours. The authors should explain their explanatory variables.
2. Classification 515 single cells into carcinoma (317) and non-carcinoma cells (175 immune cells + 23 stromal cells): It is not clear how this classification has been performed. The authors should specify their classification scheme.
3. Figure 3B: The authors argue that carcinoma cells from TNBC type tumours show lower cell-to-cell correlations compared to other carcinoma cells, but this should be supported by statistical analysis.
4. Exclusion of BC05 from Figure 4: The BC05 sample was excluded from Figure 4. Even though the authors mentioned this exclusion in the method section, the rationale behind this should be clearly mentioned in the result section to avoid any confusion.
5. Line 263, page 8: Geneset \diamond Gene set
6. Figure 6C: What does the x-axis mean? "Gene expression" is ambiguous.

Reviewer #2 (Remarks to the Author):

Authors have done a great job increasing the number of patients.

Message is clear regarding heterogeneity though clinical significance for breast cancer patients is as yet unclear.

Perhaps adding if the single cells had diverse expression of various prognostic gene signatures would be important- Recurrence Score and Oncotype and Ki67 can be easily calculated and I would expect would be different in each tumor.

It would be important to demonstrate this per subtype.

The only other comment I have is regarding batch effects. The authors say in the first results paragraph that the correlation between single cell and bulk RNASeq is poor. The PCA plot in Fig1A shows what I think are significant batch effects, which is not unexpected. The bulk samples are clearly separate from the single cells along the first principle component. The linear spreading of single cell data for each sample along PC1 also implies that batch effects or technical artefacts are present. A similar problem is seen in Fig 2A where the bulk TNBC samples all cluster together rather than their respective single cell results. This does not invalidate the rest of the analyses, but were any steps taken to assess or correct for bias?

Also note a few imprecise statements in the introduction:

Page 4, line 121, exhausted T-cells don't suppress antitumor immune effector cells, they are suppressed immune effector cells.

Statements on line 123 and 124 need references.

Reviewer #3 (Remarks to the Author):

All my concerns have been addressed by the authors.

Response to the Reviewers' comments:

Reviewer #1 (Remarks to the Author):

The revised manuscript addressed most of my concerns except some minor points.

Minor points:

1. Supplementary Fig. 2E: The description regarding the multiple regression analysis is still confusing. I think the expression levels of each single cell of a gene were used as explanatory variables to predict the expression level of bulk tumours. However, most readers would expect the regression analysis between the average of single cells and bulk tumours. The authors should explain their explanatory variables.

We agree with the reviewer that using individual cells as the explanatory components may confuse readers. Following the reviewer's suggestion, we explained that expression levels of each single cell were used as the explanatory variable in multiple regression analysis. We put this description in the figure legend.

(Figure legend 2e)

"Multiple regression analysis was performed using expression levels of each single cell as the explanatory variable to predict the expression level of bulk tumors. Adjusted R-squares of multiple regression analysis were calculated by random sampling of single cells with 1,000 iterations."

2. Classification 515 single cells into carcinoma (317) and non-carcinoma cells (175 immune cells + 23 stromal cells): It is not clear how this classification has been performed. The authors should specify their classification scheme.

To clarify the scheme of cell classification, we added a flowchart as **Fig 2a**. Subsequently the rest of figure2 labels were pushed out. (Fig 2a to Fig 2b, Fig 2b to Fig 2c, etc)

3. Figure 3B: The authors argue that carcinoma cells from TNBC type tumours show lower cell-to-cell correlations compared to other carcinoma cells, but this should be supported by statistical analysis.

Statistical analysis between each subtype is not plausible due to the small number of patients. Thus we ranked the mean value of correlation coefficient for each patient (**revised Fig. 3b, right side**) and toned down the statement.

Page 7, line 35-38:

"Carcinoma cells from TNBC type tumors (BC07-11) tend to show low cell-to-cell correlations with or without the removal of non-carcinoma cells (**Fig. 3b**), suggesting the contribution of both tumor intrinsic and microenvironmental properties to the intratumoral heterogeneity in TNBC."

4. Exclusion of BC05 from Figure 4: The BC05 sample was excluded from Figure 4. Even though the authors mentioned this exclusion in the method section, the rationale behind this should be clearly mentioned in the result section to avoid any confusion.

Following the reviewer's suggestion, we stated the exclusion in the result section.

Page 7, line 43-44:

“For the molecular subgroup analysis, we excluded BC05 tumor cells which had been subjected to neoadjuvant chemotherapy and Herceptin.”

5. Line 263, page 8: Geneset \diamond Gene set

Corrected.

6. Figure 6C: What does the x-axis mean? “Gene expression” is ambiguous.

We changed the label to “Gene expression, $\text{Log}_2(\text{TPM}+1)$ ”.

Reviewer #2 (Remarks to the Author):

Authors have done a great job increasing the number of patients.

Message is clear regarding heterogeneity though clinical significance for breast cancer patients is as yet unclear.

Perhaps adding if the single cells had diverse expression of various prognostic gene signatures would be important- Recurrence Score and Oncotype and Ki67 can be easily calculated and I would expect would be different in each tumor.

It would be important to demonstrate this per subtype.

Following the reviewer’s suggestion, we examined prognostic gene expression signatures in single cells. As Oncotype DX score was designed for the risk assessment of early stage, estrogen-receptor positive breast cancer, we used MammaPrint 70 genes (Clinical application of the 70-gene profile: the MINDACT trial, Journal of clinical oncology 26, 729-735) to calculate recurrence score. We also examined the proliferation signature with a Hallmark gene set (MSigDB) including Ki67. The results are presented in the revised supplementary figure 5. Individual cells show various distribution patterns for the proliferation and recurrence score (Supp. Fig 5A 4th and 5th rows), but also demonstrate subtype-specific trends. Overall, luminal B tumor cells have highest proliferation signature, whereas HER2 and TNBC tumor cells have highest recurrence score (Supp. Fig 5B 4th and 5th columns).

The only other comment I have is regarding batch effects. The authors say in the first results paragraph that the correlation between single cell and bulk RNASeq is poor. The PCA plot in Fig1A shows what I think are significant batch effects, which is not unexpected. The bulk samples are clearly separate from the single cells along the first principle component. The linear spreading of single cell data for each sample along PC1 also implies that batch effects or technical artefacts are present. A similar problem is seen in Fig 2A where the bulk TNBC samples all cluster together rather than their respective single cell results. This does not invalidate the rest of the analyses, but were any steps taken to assess or correct for bias?

As the reviewer commented, batch effect in single cell analysis on multiple patients/host is an important issue which should be addressed to make comparisons between patient groups.

However single cell capture-amplification process cannot be repeated with exactly same composition of single cells, which makes it difficult to separate technical batch effect from the biological variation. There are methods such as using unique molecular identifiers (UMI) instead of sequencing readcounts or using ERCC spike-ins for readcount normalization. Our data were produced without UMI or ERCC spike-ins, thus we could not apply those methods. As you will find in the method section, we used TPM (transcripts per million) values for expression quantification which were used for downstream analysis. Most of downstream analyses include mean-centering steps for normalization.

Please see below for the reviewer's only figure on the distribution of our dataset in PCA plots. In A, PC1 component shows correlation with the number of detected genes which separates the bulk from the single cells. Both PC1 and PC2 components are mainly composed of immune genes which contribute to the separation of immune cells from the tumor cells. We think this is a real biology. In B, we present 3 sets of samples in the same PCA plot, which include two separate runs on BC09 samples, primary and lymph node runs for BC03/BC03LN and for BC07/BC07LN. The pairs are not the exact replicate samples, yet mixed distribution demonstrates the technical variability between C1 runs are small.

Also note a few imprecise statements in the introduction:

Page 4, line 121, exhausted T-cells don't suppress antitumor immune effector cells, they are suppressed immune effector cells.

Statements on line 123 and 124 need references.

We changed the sentence to "Furthermore, T cells with regulatory or exhausted phenotype are associated with failure in antitumor immunity."

We slightly changed line123-124 and added references.

"A subset of B cells was proposed to promote tumor progression by affecting diverse cell types including T cells and TAMs¹⁴. However, the presence of a large number of B cells in the tumor region is associated with a good prognosis¹⁵."

Reviewer #3 (Remarks to the Author):

All my concerns have been addressed by the authors.

REVIEWERS' COMMENTS:

Reviewer #1 (Remarks to the Author):

The authors have addressed all of my concerns.

Reviewer #2: also reports to the editor that their comments have been addressed.